# Efficient Large Language Model Inference with Neural Block Linearization

**Mete Erdogan,    Francesco Tonin,    Volkan Cevher**

Laboratory for Information and Inference Systems

École Polytechnique Fédérale de Lausanne (EPFL), Switzerland

## Abstract

The high inference demands of transformer-based Large Language Models (LLMs) pose substantial challenges in their deployment. To this end, we introduce *Neural Block Linearization* (NBL), a novel framework for accelerating transformer model inference by replacing self-attention layers with linear approximations derived from Linear Minimum Mean Squared Error estimators. NBL leverages Canonical Correlation Analysis to compute a theoretical upper bound on the approximation error. Then, we use this bound as a criterion for substitution, selecting the LLM layers with the lowest linearization error. NBL can be efficiently applied to pretrained LLMs without the need for fine-tuning. In experiments, NBL achieves notable computational speed-ups while preserving competitive accuracy on multiple reasoning benchmarks. For instance, applying NBL to 12 self-attention layers in `DeepSeek-R1-Distill-Llama-8B` increases the inference speed by 32% with less than 1% accuracy trade-off, making it a flexible and promising solution to improve the inference efficiency of LLMs. The implementation is available at: https://github.com/LIONS-EPFL/NBL.

## 1 Introduction

Transformer-based models have become foundational in machine learning, with wide applications in NLP and language modeling [Vaswani et al., 2017, Brown et al., 2020, Jiang et al., 2023, Achiam et al., 2023, Dubey et al., 2024]. Due to their ability to learn long-range dependencies and capture complex patterns, transformer models have achieved state-of-the-art performance in tasks like language modeling, text generation, and translation. However, their growing size and complexity strongly limit the widespread adoption of Large Language Models (LLMs) in real-world applications, especially in resource or cost constrained scenarios.

It is therefore crucial to develop methods to reduce LLM inference costs. Proposed techniques mainly include weight pruning [Kusupati et al., 2020, Hoefler et al., 2021], low rank approximations [Yuan et al., 2023, Wang et al., 2024a], quantization [Lin et al., 2024, Saha et al., 2024], speculative decoding [Leviathan et al., 2023, Cai et al., 2024], distillation [Jiao et al., 2020, Liu et al., 2024] and subquadratic attention [Wang et al., 2024b, Zhang et al., 2025]. In this work, we focus on *pruning*, even though our method can be integrated on top of other techniques such as quantization.

Structured pruning methods eliminate specific layers, attention heads, or hidden dimensions to accelerate LLM inference [Voita et al., 2019, Ma et al., 2023, Xia et al., 2024a, Muralidharan et al., 2024, Men et al., 2024, Ashkboos et al., 2024, Song et al., 2024]. In fact, it is well-established that attention mechanisms exhibit redundancy [Voita et al., 2019, He et al., 2024]. However, many existing methods show substantial performance degradation as removing specific layers without

---

Correspondence to: Mete Erdogan <merdogan@stanford.edu>, Francesco Tonin <francesco.tonin@epfl.ch>.

39th Conference on Neural Information Processing Systems (NeurIPS 2025).

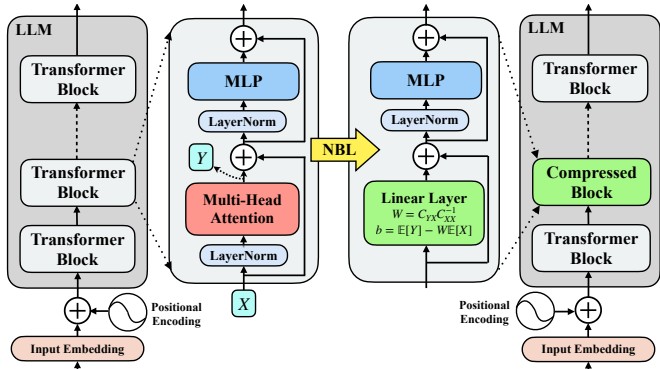

Figure 1: Illustration of Neural Block Linearization (NBL), which replaces a multi-head attention layer with an efficient linear layer using the closed-form *LMMSE* estimator.

properly replacing them can lead to substantial accuracy drop, highlighting the need for more reliable strategies to identify and replace network blocks effectively.

In this paper, we introduce Neural Block Linearization (NBL), a novel method for compressing transformer-based models by replacing self-attentions (costly network blocks) with efficient linear layers derived using Linear Minimum Mean Squared Error (*LMMSE*) estimators [Kay, 1993, Kailath et al., 2000], without the need for computationally expensive gradient-based training. While applicable to any network block, this paper mainly focuses on substituting the attention layers rather than entire transformer blocks, as experimental results show that this selective approach effectively balances inference speed-up with accuracy.

Unlike methods that entirely remove specific network components, for example Ashkboos et al. [2024], Xia et al. [2024a], Song et al. [2024], He et al. [2024], NBL maintains consistency by substituting these layers with their linear approximations. NBL further integrates theoretical linearization error quantification using Canonical Correlation Analysis (CCA) [Hotelling, 1992]. By deriving an error bound and employing it as a criterion for layer substitution, our approach provides a theoretically grounded framework to quantify redundancy, ensuring optimal compatibility with the linearization process. This enables significant inference speed-ups while maintaining performance, positioning NBL as an effective solution for optimizing large language model inference, enhancing both efficiency and scalability in applications.

**Contributions.**  Our contributions include:

- A principled and efficient substitution of attention layers with linear transformations using the closed form solution of the Linear Minimum Mean Squared Error (*LMMSE*) estimators;
- A theoretical error bound derived using Canonical Correlation Analysis (CCA), used to quantify approximation error and serve as the layer substitution criterion;
- Accelerating pre-trained transformer model inference, demonstrated through empirical results to preserve competitive performance on reasoning tasks.

## 2   Motivation

The computational efficiency of transformer-based models has become a critical concern as their use expands across NLP tasks. The attention mechanism, a cornerstone of transformer architecture, is particularly resource-intensive due to its quadratic complexity with respect to sequence length. Some methods address this issue with subquadratic linear attention mechanisms [Katharopoulos et al., 2020], e.g., [Mercat et al., 2024, Zhang et al., 2025]. These methods fundamentally change the architecture, often requiring significant retraining and leading to potential trade-offs in performance. Moreover, methods such as SLEB [Song et al., 2024], DROP [He et al., 2024] rely on redundancy quantification metrics to remove specific transformer blocks or attention layers, to accelerate inference. However, these approaches often lead to performance degradation due to the abrupt removal of critical components, which can disrupt the model's ability to capture essential patterns and dependencies.

In this work, we propose a fundamentally different approach. Rather than completely removing attention layers, we approximate those that exhibit redundancy by replacing them with linear approximations based on their input-output relationships. To achieve this, we use Canonical Correlation Analysis (CCA) [Hotelling, 1992] as a theoretical foundation to identify and quantify redundancy in a layer considering its inputs and outputs.

CCA is a statistical method that measures the linear relationship between two random vectors $X$ and $Y$. It achieves this by finding pairs of canonical directions—linear projections of $X$ and $Y$—that are maximally correlated. Mathematically, CCA solves the following optimization problem:

$$\max_{a,b} \rho = \frac{a^\top C_{YX} b}{\sqrt{a^\top C_{XX} a}\sqrt{b^\top C_{YY} b}},$$

where $C_{XX}$ and $C_{YY}$ are the covariance matrices of $X$ and $Y$, respectively, and $C_{YX}$ is their cross-covariance matrix. To compute $\rho$ and the corresponding canonical directions $a$ and $b$, we first standardize $X$ and $Y$ by normalizing their variances. This leads to the construction of the standardized cross-correlation matrix:

$$C_{\mathbb{W}} = C_{XX}^{-1/2} C_{YX} C_{YY}^{-1/2}.$$

The canonical correlations $\rho_i$ are obtained as singular values of $C_{\mathbb{W}}$ through Singular Value Decomposition (SVD):

$$C_{\mathbb{W}} = U\Sigma V^\top,$$

where $\Sigma$ is a diagonal matrix containing the singular values $\rho_i$, and $U$ and $V$ are the canonical direction matrices for $X$ and $Y$, respectively. The strength of these correlations $\rho_i$ provides a clear indication of how well the components of $Y$ can be linearly predicted from $X$. If the canonical correlations are high, it suggests that a significant portion of the output can be captured using a linear transformation, without incurring substantial loss in predictive fidelity.

We use CCA to guide the Linear Minimum Mean Squared Error (*LMMSE*) [Kay, 1993, Kailath et al., 2000] approximation of the attention layers, allowing us to quantify the approximation error introduced during linearization. By identifying and targeting the layers where the redundancy is high (i.e., canonical correlations are close to 1), our method compresses transformer models in a principled manner while preserving their functionality with minimal compromise.

Most notably, our approach replaces attention layers with linear approximations ***without any fine-tuning or gradient-based optimization, either during or after substitution***, making it highly efficient for compressing pre-trained models. Despite this, NBL achieves strong accuracy, demonstrating that attention layers can be effectively approximated without retraining. This direct substitution significantly reduces computational overhead while preserving model functionality. Our method not only enables efficient transformer compression but also provides new insights into model component analysis and redundancy quantification, aligning with the growing demand for scalable NLP solutions.

## 3   Methods

We propose a method that replaces selected blocks, particularly attention layers, with linear approximations based on their input-output relationships. This approach frames the problem as a channel estimation task, where the goal is to approximate the output of an attention layer using its input through the well-known Linear Minimum Mean Squared Error (*LMMSE*) formulation.

The *LMMSE* framework offers a principled way to minimize the mean squared error (MSE) between the true output and the estimated output of an attention layer. Using the covariance structure of the inputs and outputs, we derive the optimal linear approximation of the attention layer's behavior. Our process ensures that the replaced layer closely approximates the behavior of the original while significantly reducing computational complexity.

The following sections detail the theoretical formulation, algorithmic implementation, and practical considerations of the proposed Neural Block Linearization (NBL) method. In Section 3.1, we introduce NBL primarily in the context of replacing self-attention layers. However, NBL is a flexible framework that can be applied to any neural network block. In Section (3.2), we derive an error bound for the linearization in the NBL method, which serves as a valuable tool for redundancy characterization within the NBL framework.

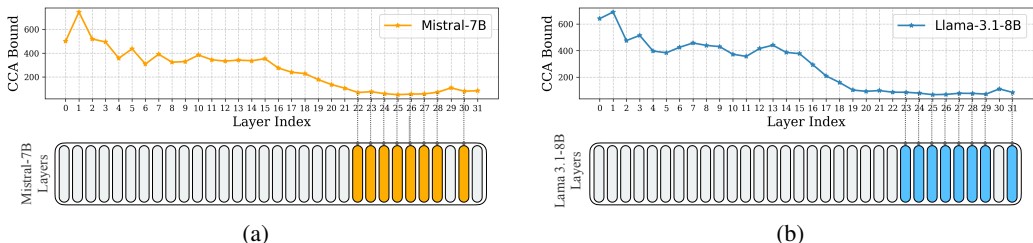

Figure 2: Illustration of layer selection in the NBL method, guided by the CCA-based bound from the Theorem 3.2, as applied to (a) `Mistral-7B` and (b) `Llama-3.1-8B` models.

## 3.1 Neural Block Linearization (NBL)

To determine the linear weights, we utilize a calibration dataset $\mathcal{D} = \{S^{(i)}\}_{i=1}^{s}$, with $s$ input sequences, each having a context length of $t$. Each sequence $S^{(i)} = \{w_1^{(i)}, w_2^{(i)}, \ldots, w_t^{(i)}\}$ is processed through multiple transformer blocks before reaching the $k$-th block. The input representations to the $k$-th attention layer are obtained by applying the first $(k-1)$ transformer blocks sequentially:

$$X_k^{(i)} = f_{k-1} \circ f_{k-2} \circ \cdots \circ f_1(S^{(i)}) \in \mathbb{R}^{h_{\text{in}} \times t},$$

where each function $f_j(\cdot)$ represents the mapping applied by the $j$-th transformer block, including its self-attention and MLP layers. The corresponding output representations are then extracted directly from the self-attention layer of the $k$-th transformer block:

$$Y_k^{(i)} = \text{A}_k(X_k^{(i)}) \in \mathbb{R}^{h_{\text{out}} \times t},$$

where $\text{A}_k(\cdot)$ denotes the softmax self-attention transformation with input and output activation sizes as $h_{\text{in}}$ and $h_{\text{out}}$, respectively, inside the $k$-th transformer block. Then, to construct the dataset for learning the linear mapping, we stack the token-wise representations across all sequences as:

$$X = \left[X_k^{(1)}, X_k^{(2)}, \ldots, X_k^{(s)}\right] \in \mathbb{R}^{h_{\text{in}} \times (s \cdot t)},$$
$$Y = \left[Y_k^{(1)}, Y_k^{(2)}, \ldots, Y_k^{(s)}\right] \in \mathbb{R}^{h_{\text{out}} \times (s \cdot t)}.$$

Here, $X_k^{(i)}$ and $Y_k^{(i)}$ are the matrices containing the representations of all tokens in sequence $i$, and stacking them along the batch dimension ensures that each column in $X$ and $Y$ corresponds to a single token's representation across all sequences. Then, $X$ and $Y$ can be treated as vector-valued random variables with $s \cdot t$ realizations. Then, our goal is to minimize the MSE between the actual attention output $Y$ and the estimated output $\hat{Y}$, formulated as:

$$\text{MSE}(Y, \hat{Y}) = \mathbb{E}\left[\|Y - \hat{Y}\|_2^2\right], \tag{1}$$

where $\mathbb{E}[\cdot]$ represents the expectation, and $||\cdot||_2$ denotes the euclidean norm of a vector. The linear estimator takes the form:

$$\hat{Y} = WX + b. \tag{2}$$

Then, the optimal linear estimator minimizing the MSE in Expression (1) is derived as presented in the following proposition.

**Proposition 3.1.** *[Kay, 1993]. The Linear Minimum Mean Squared Error (LMMSE) estimator defines the optimal linear relationship between the vector-valued random variables $X$ and $Y$ by the following weight $W$ and bias $b$:*

$$W = C_{YX}C_{XX}^{-1}, \tag{3}$$
$$b = \mathbb{E}[Y] - W\mathbb{E}[X], \tag{4}$$

*where $C_{XX}$ is the covariance of $X$ and $C_{YX}$ is the cross-covariance between $Y$ and $X$.*

This formulation demonstrates that the linearized block operates as a single linear layer with the weight $W$ and bias $b$. Figure 1 illustrates the process of using NBL to replace an attention layer of an LLM with its linear approximation. For the derivation of Proposition 3.1 and an in-depth discussion of MMSE estimators, refer to Appendix A.

## 3.2 Canonical Correlation Analysis (CCA) based error bound and redundancy analysis

A key aspect of the NBL is understanding and quantifying the error introduced by replacing a network block with its linear approximation. While the *LMMSE* estimator provides a principled approach to minimize the error between the true outputs and their linear estimates, it is essential to bound this error to ensure acceptable approximation and preserve the model performance.

To this end, we present an upper bound on the normalized mean squared error (NMSE) of the linear approximation in the following theorem. Leveraging Canonical Correlation Analysis (CCA), this theorem directly links the approximation error to the CCA singular values, which quantify the alignment between input and output spaces. These singular values provide valuable insights into the conditions under which linearization is most effective.

**Theorem 3.2.** *For a given self-attention layer with input $X$ and output $Y$, the Normalized Mean Squared Error (NMSE) of the LMMSE estimator $\hat{Y}$ is defined as: $NMSE(Y, \hat{Y}) = \frac{MSE(Y,\hat{Y})}{\text{Tr}(C_{YY})}$, where $C_{YY}$ is the auto-covariance matrix of $Y$. Then, the NMSE satisfies the following upper-bound:*

$$NMSE(Y, \hat{Y}) \leq (h_{out} - r) + \sum_{i=1}^{r}(1 - \rho_i^2), \tag{5}$$

*where $r = \min(h_{out}, h_{in})$, and $\rho_i$ are the canonical correlations between $X$ and $Y$.*

This theorem highlights that the NMSE is bounded by a summation term dependent on $1 - \rho_i^2$, where $\rho_i$ represents the canonical correlations between the input $X$ and output $Y$. Intuitively, the larger the canonical correlations ($\rho_i$ close to 1), the better the alignment (linear relationship) between the input and output spaces, and thus the lower the approximation error. The result also implies that layers with weaker correlations ($\rho_i$ closer to 0) are less suitable for linearization, as the error bound becomes larger. This insight enables the systematic identification of blocks where linearization will have less impact on overall model performance, providing a quantitative basis for deciding which attention layers or blocks to replace. The additive term corresponding to the undetermined mappings, $(h_{out} - r)$, is zero when $h_{out} \leq h_{in}$, a condition typically satisfied in most LLM layers such as self-attention.

We propose using this bound to rank the layers of a pre-trained transformer model by their suitability for linearization, as illustrated in Figures 2a and 2b. By targeting the layers with high canonical correlations, significant computational savings can be achieved with minimal approximation error. Although one could directly compute the NMSE, we adopt its CCA-based upper bound, which provides a variance-agnostic, modewise indicator of linear approximability with more stable empirical behavior (see Appendix C.1).

---

**Algorithm 1** Neural Block Linearization (NBL)

---

1: **Input**: Attention layer inputs $X$ and outputs $Y$ from calibration dataset $\mathcal{D}$, number of layers to linearize $m$.
2: **Initialization**: Load the pre-trained LLM, extract all attention layers $\mathcal{A} = \{A_1, A_2, \ldots, A_K\}$.
3: **for** each attention layer $A_k \in \mathcal{A}$ **do**
4:  Collect the input-output pairs $(X_k, Y_k)$ from $\mathcal{D}$ corresponding to the attention layer $A_k$.
5:  Compute the bound on the linearization NMSE (Theorem-3.2) for the attention layer $A_k$ using the canonical correlations $\rho_{ki}$ between $Y_k$ and $X_k$ s.t. $h = h_{\text{out}} = h_{\text{in}}$:

$$\text{NMSE}_k = \frac{\text{MSE}(Y_k, \hat{Y}_k)}{\text{Tr}(C_{YY})} \leq \sum_{i}^{h}(1 - \rho_{ki}^2).$$

6: **end for**
7: **Select** $m$ layers with the lowest NMSE bounds, forming $\mathcal{A}_{\text{low}} = \{A_{j_1}, A_{j_2}, \ldots, A_{j_m}\} \subset \mathcal{A}$.
8: **for** each layer $A_j \in \mathcal{A}_{\text{low}}$ **do**
9:  Replace $A_j$ with its linear approximation (linear layer) obtained by solving the *LMMSE* problem (Proposition 3.1), with weight and bias terms as :

$$W_j = C_{Y_j X_j} C_{X_j X_j}^{-1}, \; b_j = \mathbb{E}[Y_j] - W_j \mathbb{E}[X_j]$$

10: **end for**
11: **Output**: Compressed LLM with $m$ linearized attention layers.

---

Table 1: Calibration runtime scaling with model size.

| Llama-3.1 Model | 8B | 70B | 405B |
|---|---|---|---|
| **Hidden size** ($d$) | 4092 | 8192 | 16384 |
| **Hidden layers** | 32 | 80 | 126 |
| **Runtime / layer** | 26.04 s | 78.91 s | 371.79 s |
| **Total** | ~0.23 hr | ~1.75 hr | ~13.01 hr |

## 3.3 Algorithm

Our methodology, detailed in Algorithm 1, compresses an LLM by replacing $m$ of the computationally intensive attention layers with efficient linear estimators. Using a calibration dataset, we extract

Table 2: Performance of **Mistral-7B** on reasoning benchmarks across baselines, PruneNet [Sengupta et al., 2025], SliceGPT [Ashkboos et al., 2024], SLEB [Song et al., 2024], DROP [He et al., 2024], and ours (NBL). Prefill and throughput speeds are reported relative to the baseline original model.

| Method | Sparsity (%) | ARC-e (norm) | ARC-c (norm) | BoolQ | HellaSwag (norm) | MMLU (5-shot) | OBQA (norm) | PIQA (norm) | Wino-Grande | Avg. (↑) | Prefill (↑) | Through-put (↑) |
|---|---|---|---|---|---|---|---|---|---|---|---|---|
| **Baseline** | 0 | 79.4 | 54.2 | 83.8 | 81.1 | 65.3 | 43.8 | 82.1 | 74.2 | 70.2 | 1 | 1 |
| PruneNet-15% | 15 | 72.1 | 46.1 | 68.7 | 75.0 | 50.2 | 39.8 | 78.8 | 69.7 | 62.5 | 1.11 | 1.05 |
| PruneNet-25% | 25 | 64.3 | 39.9 | 62.7 | 67.1 | 31.6 | 35.0 | 73.1 | 66.0 | 55.0 | 1.20 | 1.09 |
| PruneNet-35% | 35 | 26.3 | 27.3 | 44.5 | 26.8 | 24.0 | 25.0 | 50.5 | 49.5 | 34.2 | 1.32 | 1.13 |
| SliceGPT-15% | 15 | 67.9 | 41.9 | 82.2 | 73.8 | 52.0 | 40.2 | 79.1 | 69.9 | 63.4 | 1.14 | 1.04 |
| SliceGPT-25% | 25 | 55.9 | 33.5 | 77.6 | 63.9 | 35.9 | 34.8 | 73.6 | 65.1 | 55.0 | 1.28 | 1.08 |
| SliceGPT-35% | 35 | 43.2 | 27.6 | 67.7 | 48.8 | 27.0 | 29.8 | 66.2 | 60.7 | 46.4 | 1.44 | 1.14 |
| SLEB-4 | 12.5 | 70.5 | 43.2 | 77.4 | 72.5 | 38.9 | 41.6 | 79.1 | 66.8 | 61.2 | 1.14 | 1.16 |
| SLEB-8 | 25 | 58.9 | 54.2 | 61.3 | 61.9 | 28.6 | 35.4 | 72.7 | 59.8 | 51.8 | 1.33 | 1.35 |
| SLEB-12 | 37.5 | 43.1 | 30.1 | 49.4 | 50.4 | 24.4 | 30.8 | 66.4 | 53.0 | 43.5 | 1.59 | 1.50 |
| Block DROP-4 | 12.5 | 70.8 | 47.2 | 80.4 | 75.4 | 61.4 | 40.2 | 77.9 | 71.0 | 65.5 | 1.14 | 1.16 |
| Block DROP-8 | 25 | 52.9 | 37.4 | 71.6 | 59.8 | 59.8 | 31.2 | 69.3 | 67.6 | 56.2 | 1.33 | 1.34 |
| Block DROP-12 | 37.5 | 31.7 | 31.6 | 42.8 | 27.0 | 24.2 | 30.0 | 54.1 | 57.1 | 37.3 | 1.59 | 1.50 |
| Block NBL-4 | 11.5 | 72.0 | 47.1 | 82.1 | 74.4 | 61.5 | 40.4 | 78.2 | 73.0 | 66.1 | 1.14 | 1.14 |
| Block NBL-8 | 23.1 | 58.8 | 38.8 | 82.2 | 60.6 | 58.6 | 36.2 | 71.4 | 69.0 | 59.4 | 1.32 | 1.31 |
| Block NBL-12 | 34.6 | 42.0 | 32.7 | 62.9 | 43.7 | 55.9 | 32.0 | 60.3 | 65.8 | 49.4 | 1.56 | 1.47 |
| Attn DROP-4 | 2.4 | 79.9 | 53.5 | 83.4 | 80.8 | 62.4 | 44.6 | 82.1 | 73.8 | 70.0 | 1.08 | 1.11 |
| Attn DROP-8 | 4.8 | 79.6 | 52.3 | 82.6 | 80.2 | 62.0 | 44.2 | 81.6 | 72.9 | 69.4 | 1.18 | 1.22 |
| Attn DROP-12 | 7.2 | 76.3 | 48.6 | 76.7 | 77.7 | 59.2 | 41.8 | 78.9 | 72.9 | 66.5 | 1.29 | 1.29 |
| Attn DROP-16 | 9.6 | 57.8 | 41.4 | 49.8 | 67.9 | 24.9 | 38.6 | 73.6 | 69.4 | 52.9 | 1.44 | 1.41 |
| Attn NBL-4 | 1.5 | 80.2 | 53.9 | 83.5 | 80.6 | 62.4 | 44.0 | 81.9 | 74.0 | 70.1 | 1.08 | 1.10 |
| Attn NBL-8 | 2.9 | 80.6 | 53.6 | 83.6 | 79.9 | 62.4 | 44.2 | 81.8 | 73.8 | 70.0 | 1.17 | 1.20 |
| Attn NBL-12 | 4.3 | 77.3 | 50.5 | 82.6 | 76.9 | 62.3 | 43.0 | 80.9 | 73.2 | 68.3 | 1.28 | 1.27 |
| Attn NBL-16 | 5.8 | 62.4 | 42.8 | 76.7 | 69.4 | 33.0 | 39.8 | 76.4 | 70.2 | 58.8 | 1.40 | 1.37 |

inputs and outputs from each attention layer and compute the weight matrices ($W_k$) and bias terms ($b_k$) via *LMMSE* estimation. Layers are ranked by CCA-derived NMSE bounds, and the $m$ layers with the lowest approximation error are selected for linearization. These layers are then replaced with their linear approximations, reducing complexity while preserving accuracy. The key computational steps of NBL have an overall complexity of $\mathcal{O}(d^3 + s \cdot t \cdot d^2)$, where $d$ is the embedding dimension of the attention layers, $s$ is the number of sequences, and $t$ is the context length of calibration samples. Since calibration is layer-wise, runtime scales linearly with the number of layers. We demonstrate that 8B-scale models can be compressed in under an hour, with scalability to larger models (see Table 1). While NBL effectively replaces attention layers, it can also linearize any network block, including transformer blocks, as also demonstrated next. Refer to Appendix B for CCA derivations, and Appendix D for NBL implementation and calibration details corresponding to Table 1.

## 4    Experiments

In this section, we assess the impact of Neural Block Linearization (NBL) on LLM inference performance and efficiency. Evaluation benchmarks include 5-shot performance on the MMLU task [Hendrycks et al., 2020] and 0-shot performance on ARC-easy (ARC-e), ARC-challenge (ARC-c) [Clark et al., 2018], BoolQ [Clark et al., 2019], HellaSwag [Zellers et al., 2019], OBQA [Mihaylov et al., 2018], PIQA [Bisk et al., 2020] and WinoGrande [Sakaguchi et al., 2021], following a similar evaluation as Zhang et al. [2025]. We implemented and evaluated NBL on an NVIDIA A100 GPU (80GB) using PyTorch [Paszke et al., 2019] and HuggingFace Transformers [Wolf, 2019]. Evaluation is carried out with the default parameters from the Evaluation Harness framework [Gao et al., 2024].

We compare NBL with several baseline methods, including SLEB [Song et al., 2024], SliceGPT [Ashkboos et al., 2024], PruneNet [Sengupta et al., 2025] and DROP [He et al., 2024], evaluating their performance on reasoning tasks and their improvements in latency and throughput. In the calibration of all methods, we used 256 samples from the C4 dataset [Raffel et al., 2020]. In the tables, "Attn NBL-$m$" denotes the NBL applied to $m$ attention layers, while "Attn DROP-$m$" removes $m$ attention layers with the DROP method. Similarly, "SLEB-$m$" refers to dropping $m$ transformer blocks based on the SLEB algorithm, and "SliceGPT-$d\%$" or "PruneNet-$d\%$" indicates pruning $d\%$ of the model parameters using SliceGPT or PruneNet methods. We note that PruneNet required tuning in our evaluations: the default $5e-4$ learning rate led to poor convergence, while $1e-4$ performed better.

Table 2 shows that Attn NBL outperforms other methods on the 32-layer `Mistral-7B` model [Jiang et al., 2023]. Replacing 8 layers (Attn NBL-8) retains the original performance (70.0), exceeding Attn DROP-8 (69.4), SliceGPT-25% (55.0), PruneNet-25% (55.0), and SLEB-8 (51.8). At higher compression (e.g., Attn NBL-16), the gap widens, where NBL scores 58.8 compared to 52.9

Table 3: Performance of **Llama-3.1-8B** across various methods and reasoning benchmarks.

| Method | Sparsity (%) | ARC-e (norm) | ARC-c (norm) | BoolQ | HellaSwag (norm) | MMLU (5-shot) | OBQA (norm) | PIQA (norm) | Wino-Grande | Avg. (↑) | Prefill (↑) | Through-put (↑) |
|---|---|---|---|---|---|---|---|---|---|---|---|---|
| **Baseline** | 0 | 81.2 | 53.6 | 81.9 | 78.9 | 65.1 | 45.0 | 81.4 | 74.2 | 70.2 | 1 | 1 |
| PruneNet-15% | 15.0 | 66.8 | 42.06 | 64.8 | 66.2 | 42.9 | 38.2 | 74.9 | 67.4 | 58.9 | 1.08 | 1.05 |
| PruneNet-25% | 25.0 | 55.7 | 33.5 | 61.8 | 56.8 | 25.5 | 32.8 | 69.0 | 62.6 | 49.7 | 1.19 | 1.09 |
| PruneNet-35% | 35.0 | 37.5 | 25.9 | 60.0 | 37.7 | 25.2 | 28.0 | 60.6 | 54.8 | 41.2 | 1.32 | 1.13 |
| SliceGPT-15% | 15.0 | 66.6 | 40.8 | 77.9 | 68.3 | 40.8 | 36.4 | 75.2 | 64.7 | 58.8 | 1.13 | 1.05 |
| SliceGPT-25% | 25.0 | 55.1 | 31.3 | 71.3 | 58.6 | 26.2 | 29.4 | 70.5 | 57.2 | 49.9 | 1.17 | 1.09 |
| SliceGPT-35% | 35.0 | 43.1 | 26.1 | 61.8 | 44.9 | 26.3 | 26.2 | 65.8 | 52.7 | 43.4 | 1.41 | 1.12 |
| SLEB-4 | 12.5 | 73.7 | 44.3 | 67.3 | 71.0 | 34.6 | 40.2 | 77.8 | 68.9 | 59.7 | 1.14 | 1.16 |
| SLEB-8 | 25.0 | 59.4 | 32.7 | 38.7 | 57.3 | 24.5 | 34.2 | 72.5 | 52.6 | 46.5 | 1.33 | 1.32 |
| SLEB-12 | 37.5 | 47.1 | 28.2 | 46.8 | 46.0 | 24.6 | 27.8 | 67.3 | 52.5 | 42.5 | 1.54 | 1.56 |
| Block DROP-4 | 12.5 | 71.4 | 47.6 | 70.6 | 73.7 | 61.7 | 41.4 | 77.5 | 70.2 | 64.3 | 1.13 | 1.16 |
| Block DROP-8 | 25.0 | 41.7 | 32.3 | 37.6 | 30.7 | 35.2 | 29.2 | 61.0 | 53.8 | 40.2 | 1.31 | 1.32 |
| Block DROP-12 | 37.5 | 39.5 | 30.6 | 56.0 | 35.1 | 46.1 | 29.8 | 59.5 | 54.6 | 43.9 | 1.54 | 1.55 |
| Block NBL-4 | 11.5 | 77.1 | 49.0 | 81.4 | 73.3 | 64.3 | 41.4 | 78.6 | 72.5 | 67.2 | 1.12 | 1.15 |
| Block NBL-8 | 23.1 | 66.2 | 41.9 | 62.5 | 62.6 | 62.6 | 35.6 | 73.1 | 70.2 | 59.3 | 1.29 | 1.27 |
| Block NBL-12 | 34.6 | 48.7 | 34.0 | 71.9 | 46.9 | 42.1 | 32.0 | 65.0 | 63.0 | 50.4 | 1.53 | 1.50 |
| Attn DROP-4 | 2.4 | 81.4 | 54.2 | 81.9 | 78.4 | 65.0 | 45.4 | 80.6 | 74.4 | 70.2 | 1.08 | 1.12 |
| Attn DROP-8 | 4.8 | 80.9 | 53.2 | 81.4 | 78.0 | 65.0 | 45.2 | 81.1 | 73.5 | 69.8 | 1.18 | 1.19 |
| Attn DROP-12 | 7.2 | 76.3 | 51.3 | 79.7 | 76.7 | 63.3 | 43.2 | 79.9 | 72.6 | 67.9 | 1.27 | 1.29 |
| Attn DROP-16 | 9.6 | 55.2 | 39.1 | 80.6 | 63.4 | 27.2 | 36.6 | 69.8 | 69.6 | 55.2 | 1.43 | 1.42 |
| Attn NBL-4 | 1.5 | 81.9 | 54.0 | 82.2 | 78.1 | 65.0 | 45.8 | 81.1 | 73.4 | 70.2 | 1.08 | 1.11 |
| Attn NBL-8 | 2.9 | 81.5 | 53.7 | 82.1 | 77.2 | 64.0 | 45.4 | 81.1 | 73.3 | 69.8 | 1.16 | 1.17 |
| Attn NBL-12 | 4.3 | 79.1 | 52.2 | 82.3 | 75.2 | 64.8 | 45.2 | 79.9 | 74.0 | 69.1 | 1.24 | 1.25 |
| Attn NBL-16 | 5.8 | 71.8 | 46.8 | 81.6 | 69.0 | 39.1 | 41.8 | 77.0 | 73.1 | 62.5 | 1.39 | 1.37 |

Table 4: Performance of **DeepSeek-R1-Distill-Llama-8B** across baselines and reasoning benchmarks. Prefill and throughput speeds are reported relative to the baseline original model.

| Method | Sparsity (%) | ARC-e (norm) | ARC-c (norm) | BoolQ | HellaSwag (norm) | MMLU (5-shot) | OBQA (norm) | PIQA (norm) | Wino-Grande | Avg. (↑) | Prefill (↑) | Through-put (↑) |
|---|---|---|---|---|---|---|---|---|---|---|---|---|
| **Baseline** | 0 | 66.1 | 42.3 | 82.9 | 74.3 | 55.7 | 41.8 | 77.8 | 67.7 | 63.6 | 1 | 1 |
| PruneNet-15% | 15.0 | 55.1 | 35.3 | 66.7 | 63.7 | 40.6 | 34.8 | 69.9 | 63.5 | 53.7 | 1.08 | 1.08 |
| PruneNet-25% | 25.0 | 44.3 | 30.5 | 62.4 | 52.6 | 27.7 | 30.2 | 65.6 | 58.2 | 46.4 | 1.16 | 1.15 |
| PruneNet-35% | 35.0 | 32.3 | 24.6 | 62.0 | 37.6 | 25.6 | 25.2 | 58.4 | 53.1 | 39.9 | 1.22 | 1.19 |
| SliceGPT-15% | 15.0 | 51.3 | 34.1 | 73.6 | 62.1 | 31.0 | 31.4 | 69.4 | 56.4 | 51.2 | 1.09 | 1.09 |
| SliceGPT-25% | 25.0 | 45.3 | 29.7 | 68.8 | 51.4 | 25.6 | 29.2 | 65.5 | 55.0 | 46.3 | 1.15 | 1.13 |
| SliceGPT-35% | 35.0 | 37.2 | 25.5 | 59.7 | 39.4 | 25.2 | 25.4 | 60.9 | 51.8 | 40.6 | 1.21 | 1.20 |
| SLEB-4 | 12.5 | 37.7 | 60.8 | 74.9 | 67.3 | 36.9 | 36.0 | 74.4 | 61.2 | 56.2 | 1.09 | 1.14 |
| SLEB-8 | 25.0 | 33.6 | 52.4 | 58.6 | 55.5 | 25.9 | 30.8 | 68.9 | 55.7 | 47.7 | 1.17 | 1.29 |
| SLEB-12 | 37.5 | 27.9 | 41.1 | 57.8 | 43.7 | 25.5 | 27.4 | 62.0 | 53.8 | 42.4 | 1.26 | 1.53 |
| Attn DROP-8 | 4.8 | 66.9 | 43.5 | 83.6 | 73.0 | 55.8 | 40.6 | 76.0 | 68.3 | 63.5 | 1.16 | 1.19 |
| Attn DROP-12 | 7.2 | 60.7 | 41.6 | 82.7 | 68.1 | 55.6 | 37.8 | 72.9 | 66.0 | 60.7 | 1.27 | 1.34 |
| Attn DROP-16 | 9.6 | 37.9 | 31.0 | 78.9 | 45.1 | 32.4 | 31.6 | 62.0 | 62.5 | 50.2 | 1.37 | 1.47 |
| Attn NBL-8 | 2.9 | 65.5 | 41.7 | 83.8 | 72.3 | 55.2 | 43.8 | 76.3 | 66.8 | 63.2 | 1.15 | 1.19 |
| Attn NBL-12 | 4.3 | 63.6 | 41.2 | 84.3 | 70.0 | 55.3 | 42.0 | 74.9 | 67.7 | 62.4 | 1.24 | 1.32 |
| Attn NBL-16 | 5.8 | 55.4 | 38.2 | 83.8 | 63.0 | 35.8 | 39.0 | 73.3 | 66.1 | 56.8 | 1.35 | 1.42 |

(DROP-16), 46.4 (SliceGPT-35%), 34.2 (PruneNet-35%), and 43.5 (SLEB-16). NBL also improves inference speed (up to 1.27×) while keeping performance loss under 2%. Similar trends appear for `Llama-3.1-8B` [Dubey et al., 2024] (Table 3) and `DeepSeek-R1-Distill-Llama-8B` [Guo et al., 2025] (Table 4), where NBL outperforms other methods, particularly with 12–16 layer replacements, confirming its advantages in improving accuracy–efficiency trade-off across models.

We report sparsity comparisons in Tables 2 to 5 alongside efficiency results. Sparsity is defined as the fraction of dropped or replaced blocks, excluding the always-preserved embedding and projection layers (1.81%, 6.54%, and 2.98% of parameters in `Mistral-7B`, `Llama-3.1-8B`, and `Llama-3.1-70B` respectively) While most parameters lie in MLPs ( 80%), attention layers dominate FLOPs and runtime, making them the most effective targets. In this view, Block NBL achieves the best accuracy–sparsity trade-off: for instance; on `Llama-3.1-8B`; Block NBL-12 attains 50.4 accuracy at 34.6% sparsity, surpassing Block DROP-8 (40.2), SLEB-8 (46.5), and SliceGPT-25% (49.9) each with a 25% sparsity. Attn NBL, while inducing less sparsity, yields the strongest accuracy–throughput gains, improving throughput by 32% on DeepSeek, with under 1% accuracy loss.

## 4.1 Latency and throughput analysis

In addition to reasoning performance, we evaluate the inference speed-up of LLMs in two stages: prompt processing (prefill) and token generation (throughput). Prompt processing builds the

Table 5: **Larger 70B models**. Performance of **Llama-3.1-*70B*** across various methods and reasoning benchmarks. Prefill and throughput speeds are reported relative to the baseline quantized model.

| Method | Sparsity (%) | ARC-e (norm) | ARC-c (norm) | BoolQ | HellaSwag (norm) | MMLU (5-shot) | OBQA (norm) | PIQA (norm) | Wino-Grande | Avg. (↑) | Prefill (↑) | Through-put (↑) |
|---|---|---|---|---|---|---|---|---|---|---|---|---|
| **Baseline** (quant.) | 0 | 86.1 | 63.7 | 84.5 | 84.6 | 77.7 | 47.8 | 84.1 | 79.2 | 76.0 | 1 | 1 |
| SLEB-16 | 20.0 | 78.7 | 55.0 | 78.3 | 77.5 | 71.2 | 44.6 | 81.1 | 77.1 | 70.4 | 1.23 | 1.24 |
| SLEB-32 | 40.0 | 60.7 | 37.2 | 64.6 | 63.6 | 44.6 | 35.8 | 73.2 | 67.3 | 55.9 | 1.63 | 1.61 |
| Attn DROP-32 | 7.1 | 85.5 | 64.1 | 84.3 | 83.7 | 77.3 | 48.0 | 84.3 | 79.0 | 75.7 | 1.20 | 1.30 |
| Attn DROP-48 | 10.1 | 77.2 | 56.2 | 82.1 | 76.3 | 70.3 | 45.6 | 79.4 | 75.1 | 70.4 | 1.29 | 1.42 |
| Attn DROP-54 | 11.9 | 43.4 | 31.4 | 65.5 | 52.2 | 36.6 | 29.4 | 67.0 | 61.0 | 48.3 | 1.34 | 1.53 |
| Attn NBL-32 | 3.9 | 86.5 | 66.4 | 85.0 | 83.2 | 77.6 | 46.8 | 83.7 | 78.8 | 76.0 | 1.16 | 1.24 |
| Attn NBL-48 | 5.9 | 79.1 | 57.7 | 83.8 | 76.2 | 71.5 | 45.0 | 81.9 | 75.3 | 71.3 | 1.25 | 1.35 |
| Attn NBL-54 | 6.6 | 63.0 | 48.2 | 82.3 | 71.6 | 65.3 | 43.4 | 78.3 | 71.7 | 65.4 | 1.30 | 1.44 |

key-value (KV) cache for a 2048-token input, while token generation autoregressively produces 2048 tokens with a batch size of 1, following He et al. [2024]. We measure the prefill speed, calculated as the number of context tokens divided by the time to generate the first token. Whereas, the throughput is defined as the median of the number of tokens generated per second, across time intervals. Results are presented in Tables 2 and 3 for `Llama-3.1-8B` and `Mistral-7B` models respectively, which show the speed-ups achieved by methods such as SliceGPT, PruneNet, SLEB, DROP, and NBL, normalized against the baseline model. Our findings show that NBL significantly improves computational efficiency, delivering consistent prefill and throughput gains across various compression levels with significantly less performance degradation than the layer-dropping methods.

## 4.2 Inference complexity and KV-cache usage

Let $K$ be the total number of attention layers in an LLM, with $m$ layers modified by NBL. The prefill speed for full attention is limited by complexity $\mathcal{O}(K \cdot n^2 d)$, where $n$ is the context length and $d$ the model dimension. Replacing $m$ attention layers reduces their complexity to $\mathcal{O}(m \cdot nd)$, yielding an overall complexity of $\mathcal{O}((K - m) \cdot n^2 d + m \cdot nd)$. Thus, prefill speed improves as $m$ increases, particularly for large $n$, where quadratic terms dominate. Correspondingly, Figure 3 shows that applying NBL to more layers results in even higher prefill speed-ups at longer contexts. Furthermore,

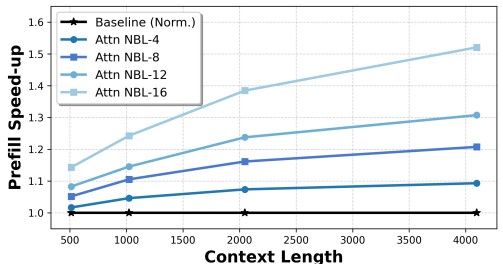

Figure 3: Prefill speed-up of **Llama-3.1-8B** with varying context lengths. NBL values are normalized by the baseline.

applying NBL reduces the usage of KV cache by a factor of $\frac{K-m}{K}$, as only the $K - m$ layers require the storage of key and value tensors. For details on the complexity, refer Appendix H.

## 4.3 Application of NBL to larger models

We extend the NBL to a larger model, namely `Llama-3.1-70B` [Dubey et al., 2024]. To fit the model in an NVIDIA A100 (80GB), we apply 4-bit post-training quantization using Activation-aware Weight Quantization (AWQ) [Lin et al., 2024] with default settings and 128 Pile samples [Gao et al., 2020] for calibration. Table 5 presents a comparison between Attn NBL and Attn DROP, with the 4-bit quantized `Llama-3.1-70B` model as baseline. For this model, PruneNet and SliceGPT were excluded from evaluation, as their public repositories currently lack support for post-quantization pruning. Both Attn NBL-32 and Attn NBL-48 outperform SLEB-16 in accuracy while matching or exceeding its inference throughput, with significantly lower KV-cache usage due to a greater number of pruned attention layers. NBL demonstrates a notable advantage by preserving model accuracy to a greater extent while achieving significant improvements in inference speed relative to baseline. For more details on quantization and implementation, see Appendix E.4.

## 4.4 Ablation studies

**Replacing the Attention layer vs Transformer block:** Tables 2 and 3 present the results for substituting $m$ transformer blocks with NBL (Block NBL-$m$) and dropping $m$ blocks using the DROP method (Block DROP-$m$). Although modifying transformer blocks achieves significant inference speed-ups, it results in much greater performance degradation compared to adjustments

made exclusively to the attention layer. Importantly, Block NBL demonstrates greater control and better accuracy preservation compared to Block DROP, as well as SliceGPT and SLEB.

**Dependency on the calibration dataset:** We examine the impact of the calibration dataset $\mathcal{D}$. During calibration, 256 samples are drawn from either the C4 or WikiText-2 training sets. The pruned models are then evaluated on both C4 and WikiText-2 validation datasets, with results provided in Tables 13 and 14, in Appendix F.1. Our comparison includes several methods, including SLEB, Attn DROP, Attn NBL (each with 8 layers dropped or linearized), and SliceGPT (with 25% sparsity). NBL performs favorably compared to other methods, where SliceGPT exhibits the greatest sensitivity.

**LoRA fine-tuning of NBL-linearized layers:** We also explore the effect of applying LoRA [Hu et al., 2022] to the linear layers produced by NBL. Experiments show that LoRA fine-tuning provides only marginal accuracy improvements over NBL alone, reinforcing the strength and generality of NBL without requiring additional tuning. Full results and setup details are provided in Appendix F.2.

**CCA-bound vs cosine distance criterion:** We further examine the impact of the layer selection criterion on NBL, comparing the CCA-bound based criterion with the cosine distance criterion originally used in the DROP method. Our findings suggest, the CCA-bound criterion provides more reliable results, likely due to its better suitability for assessing linearization error (see Appendix F.4).

## 5 Related work

The computational and memory demands of transformer-based LLMs have inspired numerous techniques to reduce the complexity of inference while maintaining the performance:

**Structured and unstructured pruning:** Techniques such as weight pruning, structured pruning, and layer removal have been widely explored to reduce model size and computational requirements. Unstructured pruning focuses on sparsifying model weights [LeCun et al., 1989, Hassibi et al., 1993, Han et al., 2015, Kusupati et al., 2020, Hoefler et al., 2021], but managing sparse matrices on modern hardware introduces significant challenges [Wang et al., 2024c]. Structured pruning methods [Voita et al., 2019, Ma et al., 2023, Xia et al., 2024a, Muralidharan et al., 2024, Men et al., 2024, Gromov et al., 2024, Song et al., 2024, He et al., 2024, Sengupta et al., 2025] target specific layers, blocks, or dimensions for removal, offering improved efficiency. However, these methods often rely on empirical insights and lack a robust theoretical framework for quantifying approximation errors.

**Low-rank approximations and quantization:** Methods leveraging low-rank decompositions [Hsu et al., 2022, Kaushal et al., 2023, Yuan et al., 2023, Wang et al., 2024a], quantization techniques [Li et al., 2024, Zhang et al., 2024a, Tseng et al., 2024], and their combinations [Yao et al., 2023, Zhang et al., 2024b, Saha et al., 2024] have been explored to simplify model inference. While NBL introduces architectural modifications by linearizing specific layers, it remains straightforward to implement on pre-trained models. It eliminates the need for data-type changes or additional hardware compatibility, streamlining its application. Also, in Section 4.3, NBL was succesfully applied to `Llama-3.1-70B`, quantized with the AWQ [Lin et al., 2024]. This highlights the potential for deeper algorithmic integration between NBL and quantization techniques for greater efficiency.

**Linear Attention mechanisms:** Linear attention mechanisms aimed at achieving sub-quadratic complexity have been proposed [Katharopoulos et al., 2020, Choromanski et al., 2020, Peng et al., 2021, Shen et al., 2021, Mercat et al., 2024, Zhang et al., 2025]. While these methods reduce inference complexity, they often demand extensive architectural modifications, retraining, or distillation, leading to trade-offs between performance and flexibility. In contrast, NBL utilizes the closed-form *LMMSE* solution to compute linear weights, avoiding these challenges.

**Speculative decoding:** Speculative decoding methods [Leviathan et al., 2023, Cai et al., 2024, Xia et al., 2024b] accelerate autoregressive generation via draft-and-verify strategies. These approaches are orthogonal to model compression techniques like NBL, as they target decoding strategy rather than model structure. ***We show that NBL can be combined with speculative decoding (e.g., the recent EAGLE-3 [Li et al., 2025]) to achieve compounding speed-ups, up to 4.07× (Table 6)*** (Please refer Appendix E.5 for more details).

Table 6: Speculative decoding + NBL speed-ups on **DeepSeek-R1-Distill-Llama-8B** (MT-bench [Zheng et al., 2023], A100).

| Model Configuration | Speedup |
|---|---|
| EAGLE-3 Alone | 3.23× |
| Attn NBL-4 + EAGLE-3 | 3.51× |
| Attn NBL-8 + EAGLE-3 | 3.89× |
| Attn NBL-12 + EAGLE-3 | 4.07× |

**Canonical Correlation Analysis (CCA) as an analysis tool:** CCA emerged as a tool to analyze and interpret neural networks. For instance, SVCCA was introduced to compare representations across layers and models [Raghu et al., 2017], and CCA was applied to investigate robustness and generalization [Morcos et al., 2018]. Additionally, a similarity index for comparing representations between different architectures was proposed [Kornblith et al., 2019], and representation alignment in wide networks was studied [Nguyen et al., 2020]. These works highlight the effectiveness of CCA in examining neural network structure and functionality. Similarly, we use CCA to analyze neural activations and assess the suitability of attention layers for linear approximations.

## 6 Further discussions, possible limitations, and future work

Empirical results show the effectiveness of our approach across several challenging reasoning benchmarks (e.g., MMLU, HellaSwag, ARC). Nonetheless, it is important to acknowledge certain limitations. In general, certain nonlinear transformations in large language models may not admit accurate linear approximations. This is an inherent property of deep architectures. NBL addresses this by ranking layers based on linear predictability via CCA, and substituting only those with low approximation error. The method is explicitly designed for such selective substitution; applying it indiscriminately to all attention layers would overly simplify the model, leading to shallow and history-agnostic behavior, which our approach does not advocate.

The choice of calibration dataset can affect performance on new inputs, where a form of calibration history averaging could lead to hallucination-like effects. These effects could be possible especially in domain shift or insufficient calibration coverage cases. Our ablations show that NBL demonstrates superior robustness than SliceGPT and that NBL's performance is remarkably stable even in the case of two very different calibration distributions (C4 and WikiText-2). Despite being calibrated on C4, the model was tested on reasoning tasks that likely exhibit substantially different data distributions, yet it achieved consistently strong performance (Tables 2–5).

We believe NBL offers significant promise for future research on optimizing LLM inference. Although NBL shows only mild sensitivity to calibration data, this can be further reduced using calibration data mixture techniques or adaptive strategies. A promising direction is to combine theoretical analysis of QKV structure with data-driven calibration to improve generalization and reduce reliance on task-specific data. More principled selection of layers could also be achieved by analysing operator norms of attention layers, or examining the internal behavior of attention matrices. Investigating the relative norms of attention outputs and residual connections may further complement the CCA based criterion. Finally, hybrid designs that retain critical attention heads within a layer present a promising path for balancing inference efficiency with contextual reasoning capacity.

## 7 Conclusion

This work proposes Neural Block Linearization (NBL), a novel framework for the compression of transformer-based LLMs by selectively replacing attention layers or other computation-intensive network blocks with linear approximations derived using Linear Minimum Mean Squared Error estimators. NBL effectively integrates theoretical rigor into practical implementation, resorting to Canonical Correlation Analysis to bound approximation errors and guide layer substitution. Experimental results demonstrate that NBL achieves significant computational efficiency while maintaining competitive performance across various reasoning benchmarks, outperforming other recent pruning techniques in most cases, without resorting to additional retraining, and adapts smoothly to pre-trained models. These directions position NBL as a scalable and adaptable solution for compressing and deploying LLMs in resource-constrained environments.

## Acknowledgements

This work was supported by Hasler Foundation Program: Hasler Responsible AI (project number 21043). This research was sponsored by the Army Research Office and was accomplished under Grant Number W911NF-24-1-0048. This work was supported by the Swiss National Science Foundation (SNSF) under grant number 200021_205011.

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

# Appendix

## Table of contents

# A  Overview of Linear Minimum Mean Squared Error (*LMMSE*) estimation

Minimum Mean Squared Error (MMSE) estimation is a fundamental statistical method for estimating an unknown quantity from noisy observations. The goal of MMSE estimation is to minimize the mean squared error (MSE) between the actual value and its estimate. While the general MMSE approach does not impose restrictions on the form of the estimator, the Linear Minimum Mean Squared Error (*LMMSE*) estimation specifically considers linear estimators, making it computationally efficient and widely applicable. This section provides a concise overview of *LMMSE* estimation, including its derivation and its connection to our proposed method. For a more detailed discussion on *LMMSE* estimation, refer to the textbooks by Kailath et al. [2000] and Kay [1993].

## A.1  General MMSE estimation

In the general MMSE framework, given observations $X \in \mathbb{R}^m$ and a quantity to be estimated $Y \in \mathbb{R}^n$, the MMSE estimator is defined as the conditional expectation of $Y$ given $X$:

$$\hat{Y} = \mathbb{E}[Y|X]. \tag{6}$$

This estimator minimizes the MSE, defined as:

$$\mathrm{MSE}(Y, \hat{Y}) = \mathbb{E}\left[\|Y - \hat{Y}\|_2^2\right]. \tag{7}$$

The MMSE estimator is optimal in terms of minimizing the expected squared error and does not assume a specific functional form for the relationship between $X$ and $Y$. However, the computation of the conditional expectation can be intractable in many practical scenarios, especially for high-dimensional or non-Gaussian distributions. This motivates the use of *LMMSE*, which simplifies the problem by restricting the estimator to a linear form.

## A.2  Linear MMSE (*LMMSE*) estimation

In the *LMMSE*, the estimator $\hat{\mathbf{y}}$ is constrained to be a linear function of the observations $\mathbf{x}$, given by:

$$\hat{Y} = WX + b, \tag{8}$$

where $W \in \mathbb{R}^{n \times m}$ is the weight matrix, and $\mathbf{b} \in \mathbb{R}^n$ is the bias vector. The goal is to determine $W$ and $b$ such that the MSE is minimized. The MSE is expressed as:

$$\begin{aligned}
\mathrm{MSE}(Y, \hat{Y}) &= \mathbb{E}\left[\|Y - \hat{Y}\|_2^2\right] \\
&= \mathrm{Tr}\left(\mathbb{E}\left[(Y - WX - b)(Y - WX - b)^T\right]\right).
\end{aligned}$$

### A.2.1  Properties of *LMMSE*

The *LMMSE* estimator has several key properties. It minimizes the MSE among all linear estimators and coincides with the general MMSE estimator when the joint distribution of $X$ and $Y$ is Gaussian. An important property of the *LMMSE* estimator is the orthogonality of the estimation error $\mathcal{E} = Y - \hat{Y}$ to the observations $X$, such that $\mathbb{E}\left[\mathcal{E}(X - \mathbb{E}[X])^T\right] = \mathbf{0}$. This orthogonality implies that no further reduction in error is possible through linear adjustments.

### A.2.2  Common Use Cases of *LMMSE*

*LMMSE* estimation is widely applied in various fields. In signal processing, it is used for tasks such as noise reduction, channel equalization, and adaptive filtering. In communications, it serves as a foundational method for channel estimation and interference cancellation. In sensor networks, it is utilized to fuse noisy measurements from multiple sensors for a more accurate state estimation. Control systems leverage *LMMSE* for state estimation, with Kalman and Wiener filtering methods being prominent examples.

In machine learning, *LMMSE* underpins fundamental techniques such as linear regression, where it estimates optimal weights by solving a least-squares problem to establish relationships between input features and outputs. Beyond its computational advantages, *LMMSE* provides a robust framework for analyzing noise-affected data. In addition, in fields such as medical imaging and remote sensing, *LMMSE* is used to denoise and reconstruct images, delivering high-quality results from noisy or incomplete measurements.

### A.2.3   Advantages and limitations of *LMMSE*

The *LMMSE* is computationally efficient due to its closed-form solution, making it suitable for real-time applications. It provides interpretable results when the covariance structures are well understood and is robust under Gaussian noise assumptions. However, reliance on accurate covariance estimation and the linearity assumption can limit its effectiveness in highly non-linear systems. Furthermore, it may be sensitive to outliers and model specifications. Despite these challenges, *LMMSE* remains an important framework in estimation theory, valued for its theoretical rigor and practical utility.

The advantages of using *LMMSE* in this work include its ability to maintain computational efficiency, as the closed-form solution avoids the need to retrain the model after layer replacement. Additionally, *LMMSE* ensures that the approximation error is minimized in a principled way, leveraging the statistical structure of the data. This property is critical for preserving the model's performance on downstream tasks after compression.

However, the linearity assumption inherent in *LMMSE* introduces limitations. While the approximation is effective for layers with strong linear correlations between inputs and outputs, it may not capture the complex, nonlinear interactions in some attention layers. This limitation is addressed in the paper by applying *LMMSE* selectively to layers where canonical correlations suggest that linearization will have minimal impact on performance.

### A.3   Derivation of the *LMMSE* estimator (Proposition 3.1)

*Proof.* To derive the Linear Minimum Mean Squared Error (*LMMSE*) estimator, we first solve for the bias vector $b$:

$$\text{MSE}(Y, \hat{Y}) = \text{Tr}\left(\mathbb{E}\left[(Y - WX - b)(Y - WX - b)^T\right]\right)$$
$$= \text{Tr}\left(\mathbb{E}[YY^T] - \mathbb{E}[Y(WX + b)^T] - \mathbb{E}[(WX + b)Y^T] + \mathbb{E}[(WX + b)(WX + b)^T]\right).$$

Expanding the quadratic term $\mathbb{E}[(WX + b)(WX + b)^T]$:

$$\mathbb{E}[(WX + b)(WX + b)^T] = \mathbb{E}[WXX^T W^T] + 2b\mathbb{E}[X]^T W^T + bb^T.$$

The derivative of the trace of a matrix $\text{Tr}(X)$ with respect to a matrix $X$ is simply the identity matrix, i.e.,

$$\frac{\partial \text{Tr}(X)}{\partial X} = I,$$

where the trace operator sums the diagonal elements of $X$. This result is used in conjunction with the chain rule in the derivation to handle the differentiation of MSE that includes trace terms with respect to $W$ and $b$. Then, the derivative of the MSE with respect to $b$ is:

$$\frac{\partial \text{MSE}(Y, \hat{Y})}{\partial b} = -2\mathbb{E}[Y] + 2b + 2W\mathbb{E}[X].$$

Setting $\frac{\partial \text{MSE}(Y, \hat{Y})}{\partial b} = 0$, we solve for $b$:

$$b = \mathbb{E}[Y] - W\mathbb{E}[X].$$

Next, substituting $b$ back into the MSE expression, we focus on the centered terms $X - \mathbb{E}[X]$ and $Y - \mathbb{E}[Y]$:

$$\text{MSE}(Y, \hat{Y}) = \text{Tr}\bigg( \mathbb{E}\left[(Y - \mathbb{E}[Y])(Y - \mathbb{E}[Y])^T\right] - \mathbb{E}\left[(Y - \mathbb{E}[Y])(W(X - \mathbb{E}[X]))^T\right].$$

$$- \mathbb{E}\left[(W(X - \mathbb{E}[X]))(Y - \mathbb{E}[Y])^T\right] + \mathbb{E}\left[(W(X - \mathbb{E}[X]))(W(X - \mathbb{E}[X]))^T\right] \bigg).$$

The derivative of the MSE with respect to $W$ is:

$$\frac{\partial \text{MSE}(Y, \hat{Y})}{\partial W} = -2\mathbb{E}[(Y - \mathbb{E}[Y])(X - \mathbb{E}[X])^T] + 2W\mathbb{E}[(X - \mathbb{E}[X])(X - \mathbb{E}[X])^T].$$

Setting $\frac{\partial \text{MSE}(Y, \hat{Y})}{\partial W} = 0$, we find:

$$W\mathbb{E}[(X - \mathbb{E}[X])(X - \mathbb{E}[X])^T] = \mathbb{E}[(Y - \mathbb{E}[Y])(X - \mathbb{E}[X])^T].$$

Rewriting in terms of covariance matrices:

$$WC_{XX} = C_{YX},$$

where $C_{XX}$ and $C_{YX}$ are:

$$C_{XX} = \mathbb{E}\left[(X - \mathbb{E}[X])(X - \mathbb{E}[X])^T\right], \quad C_{YY} = \mathbb{E}\left[(Y - \mathbb{E}[Y])(X - \mathbb{E}[X])^T\right].$$

The optimal $W$ is then:

$$W = C_{YX}C_{XX}^{-1}.$$

Substituting $W$ and $b$ into the linear model, the *LMMSE* estimator becomes:

$$\hat{Y} = \mathbb{E}[Y] + C_{YX}C_{XX}^{-1}(X - \mathbb{E}[X]).$$

This form highlights how the *LMMSE* estimator utilizes the mean and covariance of $X$ and $Y$ to minimize the MSE. $\qquad\square$

# B  Overview of Canonical Correlation Analysis (CCA)

Canonical Correlation Analysis (CCA) is a statistical technique that identifies linear transformations of two random vectors, $X \in \mathbb{R}^p$ and $Y \in \mathbb{R}^q$, such that the transformed variables are maximally correlated. The goal is to find pairs of canonical variables $u_i = a_i^T X$ and $v_i = b_i^T Y$ for $i \in \{1, \ldots, \min(p, q)\}$, where $a_i \in \mathbb{R}^p$ and $b_i \in \mathbb{R}^q$, with the correlation $\rho_i$ between $u_i$ and $v_i$ being maximized.

## B.1  Problem formulation

CCA aims to solve the following optimization problem:

$$\max_{a,b} \rho = \frac{a^T C_{XY} b}{\sqrt{a^T C_{XX} a}\sqrt{b^T C_{YY} b}},$$

where $C_{XX}, C_{YY}, C_{XY}$ are the autocovariance matrices of $X, Y$, and their cross-covariance, respectively.

## B.2  Canonical Correlation matrix

To simplify the problem, a change of basis is applied to normalize the covariance matrices $C_{XX}$ and $C_{YY}$. Define:

$$\tilde{x}_1 = C_{XX}^{1/2} a, \quad \tilde{y}_1 = C_{YY}^{1/2} b,$$

where the transformed variables $\tilde{x}_1$ and $\tilde{y}_1$ have unit variance. Substituting these into the optimization problem and using the Cauchy-Schwarz inequality, the problem reduces to an eigenvalue problem:

$$\tilde{x}_1 = \arg\max_{\tilde{x}} \frac{\tilde{x}^T (C_{XX}^{-\frac{1}{2}} C_{XY} C_{YY}^{-1} C_{YX} C_{XX}^{-\frac{1}{2}}) \tilde{x}}{\|\tilde{x}\|}.$$

The solution involves solving the eigenvalue problem:

$$C_{XX}^{-\frac{1}{2}} C_{XY} C_{YY}^{-1} C_{YX} C_{XX}^{-\frac{1}{2}} \tilde{x} = \lambda \tilde{x}, \tag{9}$$

where the eigenvalues $\lambda_i$ are the squared canonical correlations $\rho_i^2$, and the corresponding eigenvectors $\tilde{x}_i$ determine the canonical weights in the whitened space.

## B.3  Performing CCA using the Singular Value Decomposition (SVD)

The eigenvalue problem in (9) can alternatively be solved by performing the full SVD on the matrix

$$A = C_{XX}^{-\frac{1}{2}} C_{XY} C_{YY}^{-\frac{1}{2}} = U \Sigma V^T,$$

where $U \in \mathbb{R}^{p \times p}$ and $V \in \mathbb{R}^{q \times q}$ are orthogonal matrices, and $\Sigma \in \mathbb{R}^{p \times q}$ is a diagonal matrix containing the singular values. From this decomposition, the relationship $AA^T = U\Sigma^2 U^T$ is obtained. Then, the matrix in the eigenvalue problem in (9), given by $C_{XX}^{-\frac{1}{2}} C_{XY} C_{YY}^{-1} C_{YX} C_{XX}^{-\frac{1}{2}}$ is symmetric and therefore orthogonally diagonalizable as:

$$C_{XX}^{-\frac{1}{2}} C_{XY} C_{YY}^{-1} C_{YX} C_{XX}^{-\frac{1}{2}} = Q\Lambda Q^T,$$

where $Q \in \mathbb{R}^{p \times p}$ is an orthogonal matrix and $\Lambda \in \mathbb{R}^{p \times p}$ is a diagonal matrix containing the eigenvalues. Noting that $AA^T = C_{XX}^{-\frac{1}{2}} C_{XY} C_{YY}^{-1} C_{YX} C_{XX}^{-\frac{1}{2}}$, we can equate:

$$Q\Lambda Q^T = U\Sigma^2 U^T.$$

Since $Q$ and $U$ are orthogonal matrices and diagonalize the same symmetric matrix, it follows that $\Lambda = \Sigma^2$. Thus, performing SVD on $C_{XX}^{-\frac{1}{2}} C_{XY} C_{YY}^{-\frac{1}{2}}$ provides the canonical correlations $\rho_1, \rho_2, \ldots$ as the singular values of $\Sigma$, and the results for CCA can be summarized as follows:

- $U \in \mathbb{R}^{p \times p}$: Orthogonal matrix containing the left singular vectors, which correspond to the canonical weights for $X$,

- $V \in \mathbb{R}^{q \times q}$: Orthogonal matrix containing the right singular vectors, which correspond to the canonical weights for $Y$,
- $\Sigma \in \mathbb{R}^{p \times q}$: Diagonal matrix containing the canonical correlations $\rho_1, \rho_2, \ldots \rho_{min(p,q)}$, which are the square roots of the eigenvalues $\lambda_i$ in $\Lambda$.

Then, the canonical variables can also be computed as:

$$u_i = a_i^T X = (C_{XX}^{-1/2} U_i)^T X, \tag{10}$$

$$v_i = b_i^T Y = (C_{YY}^{-1/2} V_i)^T Y, \tag{11}$$

where $U_i$ and $V_i$ are the $i$-th columns of $U$ and $V$, respectively. Furthermore, the singular values in $\Sigma$ are the canonical correlations $\rho_i$. If $\rho_i = 1$ ($\rho_i = 0$), the $i$-th canonical pair is perfectly correlated (uncorrelated).

# C   Proof of Theorem 3.2

*Proof.* First, if we put the description of the linear estimator $\hat{Y}$ as in (1) into the MSE formula in (2) we have:

$$\begin{aligned}
\text{MSE}(Y, \hat{Y}) &= \mathbb{E}\left[\|Y - \hat{Y}\|_2^2\right]. \\
&= \text{Tr}\left(\mathbb{E}\left[(Y - \hat{Y})(Y - \hat{Y})^T\right]\right) \\
&= \text{Tr}\left(\mathbb{E}\left[(Y - WX - b)(Y - WX - b)^T\right]\right) \\
&= \text{Tr}\left(\mathbb{E}\left[(Y - WX - \mathbb{E}[Y] + W\mathbb{E}[X])(Y - WX - \mathbb{E}[Y] + W\mathbb{E}[X])^T\right]\right) \\
&= \text{Tr}\left(\mathbb{E}\left[((Y - \mathbb{E}[Y]) - W(X - \mathbb{E}[X]))(Y - \mathbb{E}[Y]) - W(X - \mathbb{E}[X])^T\right]\right),
\end{aligned}$$

where $\text{Tr}(\cdot)$ is the trace operator defined on matrices. Then, opening the outer-product, we end up with the following expression:

$$\begin{aligned}
\text{MSE}(Y, \hat{Y}) = \text{Tr}\Big(&\mathbb{E}\left[(Y - \mathbb{E}[Y])(Y - \mathbb{E}[Y])^T\right] \\
&- \mathbb{E}\left[(Y - \mathbb{E}[Y])(X - \mathbb{E}[X])^T W^T\right] \\
&- \mathbb{E}\left[W(X - \mathbb{E}[X])(Y - \mathbb{E}[Y])^T\right] \\
&+ \mathbb{E}\left[W(X - \mathbb{E}[X])(X - \mathbb{E}[X])^T W^T\right].\Big)
\end{aligned}$$

As the weight matrices $W$ are deterministic, we can take them out of the expectation, and simplify the covariance expressions using $C$, and using the fact $C_{XY}^T = C_{YX}$ in the second line below:

$$\begin{aligned}
\text{MSE}(Y, \hat{Y}) &= \text{Tr}\left[C_{YY} - C_{YX}W^T - WC_{XY} + WC_{XX}W^T\right] \\
&= \text{Tr}\left[C_{YY} - C_{YX}(C_{YX}C_{XX}^{-1})^T - C_{YX}C_{XX}^{-1}C_{XY} + C_{YX}C_{XX}^{-1}C_{XX}(C_{YX}C_{XX}^{-1})^T\right] \\
&= \text{Tr}\left[C_{YY} - C_{YX}C_{XX}^{-1}C_{XY} - C_{YX}C_{XX}^{-1}C_{XY} + C_{YX}C_{XX}^{-1}\cancel{C_{XX}C_{XX}^{-1}}C_{XY}\right] \\
&= \text{Tr}\left[C_{YY} - C_{YX}C_{XX}^{-1}C_{XY} - \cancel{C_{YX}C_{XX}^{-1}C_{XY}} + \cancel{C_{YX}C_{XX}^{-1}C_{XY}}\right] \\
&= \text{Tr}\left[C_{YY} - C_{YX}C_{XX}^{-1}C_{XY}\right]. \tag{12}
\end{aligned}$$

Then, by using the matrix square root of $C_{YY}$, and using the fact that for $A \in \mathbb{R}^{K \times L}$ and $B \in \mathbb{R}^{L \times K}$, $\text{Tr}(AB) = \text{Tr}(BA)$, we can further simplify the expression:

$$\begin{aligned}
\text{MSE}(Y, \hat{Y}) &= \text{Tr}\left[C_{YY} - C_{YX}C_{XX}^{-1}C_{XY}\right] \\
&= \text{Tr}\left[C_{YY}^{\frac{1}{2}}\left[\mathbb{I} - C_{YY}^{-\frac{1}{2}}C_{YX}C_{XX}^{-1}C_{XY}C_{YY}^{-\frac{1}{2}}\right]C_{YY}^{\frac{1}{2}}\right] \\
&= \text{Tr}\left[\left[\mathbb{I} - C_{YY}^{-\frac{1}{2}}C_{YX}C_{XX}^{-1}C_{XY}C_{YY}^{-\frac{1}{2}}\right]C_{YY}\right] \\
&= \text{Tr}\left[\left[\mathbb{I} - C_{YY}^{-\frac{1}{2}}C_{YX}C_{XX}^{-\frac{1}{2}}C_{XX}^{-\frac{1}{2}}C_{XY}C_{YY}^{-\frac{1}{2}}\right]C_{YY}\right] \\
&= \text{Tr}\left[\left[\mathbb{I} - C_{YY}^{-\frac{1}{2}}C_{YX}C_{XX}^{-\frac{1}{2}}\left(C_{YY}^{-\frac{1}{2}}C_{YX}C_{XX}^{-\frac{1}{2}}\right)^T\right]C_{YY}\right]
\end{aligned}$$

Here, the canonical correlation matrix is $\text{corr}(Y, X) = C_{YY}^{-\frac{1}{2}} C_{YX} C_{XX}^{-\frac{1}{2}}$. Then if we apply full Singular Value Decomposition on this matrix as $U\Sigma V^T = \text{corr}(Y, X)$ such that $U \in \mathbb{R}^{h_{\text{out}} \times h_{\text{out}}}$, $\Sigma \in \mathbb{R}^{h_{\text{out}} \times h_{\text{in}}}$ and $V \in \mathbb{R}^{h_{\text{in}} \times h_{\text{in}}}$. Also, using the fact that $U$ and $V$ are orthogonal matrices such that $UU^T = U^T U = \mathbb{I}$ and $VV^T = V^T V = \mathbb{I}$, we have:

$$
\begin{aligned}
\text{MSE}(Y, \hat{Y}) &= \text{Tr}\left[\left[\mathbb{I} - U\Sigma V^T \left(U\Sigma V^T\right)^T\right] C_{YY}\right] \\
&= \text{Tr}\left[\left[\mathbb{I} - U\Sigma\Sigma^T U^T\right] C_{YY}\right] \\
&= \text{Tr}\left[\left[UU^T - U\Sigma\Sigma^T U^T\right] C_{YY}\right] \\
&= \text{Tr}\left[U\left[\mathbb{I} - \Sigma\Sigma^T\right] U^T C_{YY}\right] \\
&= \text{Tr}\left[\left[\mathbb{I} - \Sigma\Sigma^T\right] U^T C_{YY} U\right].
\end{aligned}
$$

We have $\Sigma\Sigma^T = diag(\rho_1^2, \rho_2^2, \ldots, \rho_{h_{\text{out}}}^2)$ where $\rho_i = 0$ for $i > r = min(h_{\text{out}}, h_{\text{in}})$. Then, we can write the MSE expression as:

$$
\begin{aligned}
\text{MSE}(Y, \hat{Y}) &= \text{Tr}\left[\left(\mathbb{I} - diag(\rho_1^2, \rho_2^2, \ldots, \rho_{h_{\text{out}}}^2)\right) U^T C_{YY} U\right] \\
&= \sum_i^{h_{\text{out}}} (1 - \rho_i^2)(U^T C_{YY} U)_{ii}.
\end{aligned}
$$

Then the product $(U^T C_{YY} U)_{ii}$ can be written as the following by using the symmetric matrix singular value decomposition of the covariance matrix $C_{YY} = U_Y \Sigma_Y U_Y^T$ and $\Sigma_Y = diag(\sigma_{Y,1}, \sigma_{Y,2}, \ldots, \sigma_{Y,h_{\text{out}}})$:

$$
\begin{aligned}
(U^T C_{YY} U)_{ii} &= \sum_k (U^T U_Y \Sigma_Y)_{ik}(U_Y^T U)_{ki} \\
&= \sum_k ((U^T U_Y)_{ik} \sigma_{Y,k})(U^T U_Y)_{ik} \\
&= \sum_k \sigma_{Y,k}(U^T U_Y)_{ik}^2
\end{aligned}
$$

Then, we have the MSE expression as

$$
\begin{aligned}
\text{MSE}(Y, \hat{Y}) &= \sum_i (1 - \rho_i^2)(U^T C_{YY} U)_{ii} \\
&= \sum_i (1 - \rho_i^2) \sum_k \sigma_{Y,k}(U^T U_Y)_{ik}^2 \\
&\leq \sum_i (1 - \rho_i^2) \sum_k \sigma_{Y,k} \\
&= \sum_i (1 - \rho_i^2) Tr(C_{YY}),
\end{aligned}
$$

where the inequality arises from the fact that $U^T U_Y$ is a product of orthogonal matrices, ensuring that its elements have magnitudes less than or equal to one. Also, the last equality comes by the fact that the sum of the singular values of a matrix is equal to the trace of the same matrix. Then we can write the bound, as the MSE normalized by the trace of the auto-correlation

$$
\begin{aligned}
\text{NMSE}(Y, \hat{Y}) = \frac{\text{MSE}(Y, \hat{Y})}{Tr(C_{YY})} &\leq \sum_{i=1}^{h_{\text{out}}} (1 - \rho_i^2) \\
&= (h_{\text{out}} - r) + \sum_{i=1}^{r} (1 - \rho_i^2),
\end{aligned}
$$

such that $r = min(h_{\text{out}}, h_{\text{in}})$ using the fact $\rho_i = 0$ for $i > r$. $\qquad \square$

## C.1 Rationale for the CCA-based upper bound in layer substitutability

Let $Y \in \mathbb{R}^d$ denote a layer's output and $\widehat{Y}$ its linear approximation from the corresponding input $X$. Writing the error as $E := Y - \widehat{Y}$, the normalized mean-squared error (NMSE) is

$$\text{NMSE}(Y, \widehat{Y}) \;=\; \frac{\mathbb{E}\|Y - \widehat{Y}\|_2^2}{\mathbb{E}\|Y\|_2^2} \;=\; \frac{\text{Tr}[C_{YY} - C_{YX}C_{XX}^{-1}C_{YX}^\top]}{\text{Tr}[C_{YY}]}. \tag{13}$$

Our analysis employs a CCA-based upper bound that evaluates a *whitened* error:

$$\text{NMSE}(Y, \widehat{Y}) \leq (h_{\text{out}} - r) + \sum_{i=1}^{r}(1 - \rho_i^2) \tag{14}$$

where $\rho_1 \geq \cdots \geq \rho_r$ are the canonical correlations between $X$ and $Y$ (with $r := \text{rank}$). This connects per-mode approximation quality directly to canonical alignment.

The choice of a CCA-based bound over the scalar NMSE in (13) is motivated by the following considerations:

1. *Mode-wise, variance-agnostic assessment.* The scalar NMSE aggregates error with variance weighting through $\text{Tr}[C_{YY}]$; high-variance directions can dominate, potentially obscuring failure in low-variance (but semantically important) modes. In contrast, (14) decomposes error along canonical directions and evaluates *per-mode* alignment via $1 - \rho_i^2$, after whitening by $C_{YY}^{-1/2}$. This avoids skew from marginal variances and directly targets linear predictability of $Y$ from $X$, which is the relevant notion for safe linear substitution.

2. *Comparable computational profile with improved numerical behavior.* Both approaches require second-order statistics, i.e., estimates of $C_{XX}$, $C_{YY}$, and $C_{XY}$. Forming linear predictors and their induced error covariances for NMSE involves matrix inversions/multiplications; the CCA route performs whitening and an SVD of the whitened cross-covariance. The big-$O$ complexity is comparable, while the explicit whitening in CCA often yields more stable behavior in low-sample or near-singular regimes by regularizing anisotropy in the covariance estimates.

3. *Empirical evidence for superior layer selection.* On MISTRAL-7B, selecting the 12 most "substitutable" layers by the CCA-based criterion (smallest average $1 - \rho_i^2$) preserves reasoning accuracy significantly better than selecting by direct NMSE:

   | Criterion | Baseline (%) | After replacement (%) |
   |-----------|--------------|------------------------|
   | CCA-based | 70.2 | 68.3 |
   | Direct NMSE | 70.2 | 39.1 |

   The CCA-based selection aligns with the goal of identifying layers whose outputs are well explained by linear projections of their inputs, thereby enabling safer linear substitution.

While NMSE is a standard scalar figure of merit in estimation theory [Kay, 1993, Kailath et al., 2000], it can be biased toward high-variance directions and provides limited diagnostic insight. The CCA-based bound in (14) delivers a principled, variance-agnostic, and mode-wise indicator tightly coupled to linear substitutability. This theoretical alignment, together with comparable computational cost and favorable empirical behavior, motivates its use as the guiding criterion for layer replacement.

CCA is a classical tool for quantifying linear relations between multivariate random vectors and has been widely used to probe representational alignment and predictability [Hotelling, 1992, Hardoon et al., 2004]. Its canonical-mode decomposition naturally supports analyses like the Equation (14), that evaluate approximation quality per direction.

# D Algorithmic details for NBL

---

**Algorithm 2** Pseudocode for NBL Weight/Bias and CCA Bound Computation for a given Self-Attention Layer

---

1: **Input:** $X$ (attention layer input), $Y$ (attention layer output).
2: **Output:** Weight matrix $W$ and bias $b$, CCA-bound.
3: $Y_+ \leftarrow Y + X$   {Residual connection output.}
4:
5: **Mean Computations**
6: $E[X] \leftarrow \text{Mean}(X)$
7: $E[Y] \leftarrow \text{Mean}(Y)$
8: $E[Y_+] \leftarrow \text{Mean}(Y_+)$
9:
10: **Cross-Covariance**
11: $C_{YX} \leftarrow Cov(Y, X)$   {Unbiased estimator for cross-covariance.}
12: $C_{Y_+X} \leftarrow Cov(Y_+, X)$
13:
14: **Covariance Matrices**
15: $C_{XX} \leftarrow Cov(X, X)$   {Unbiased estimator for auto-covariance.}
16: $C_{Y_+Y_+} \leftarrow Cov(Y_+, Y_+)$
17:
18: **Eigen-Decomposition**
19: $(\lambda_{Y_+Y_+}, V_{Y_+Y_+}) \leftarrow \text{Eigh}(C_{Y_+Y_+})$   {Symmetric matrix eigenvalue decomposition.}
20: $(\lambda_{XX}, V_{XX}) \leftarrow \text{Eigh}(C_{XX})$
21:
22: **Inverse Square Roots**
23: $C_{Y_+Y_+}^{-1/2} \leftarrow V_{YY} \, \text{diag}\big(\lambda_{Y_+Y_+}^{-1/2}\big) V_{Y_+Y_+}^T$   {Matrix square roots $(C_{Y_+Y_+} = C_{Y_+Y_+}^{1/2} \ C_{Y_+Y_+}^{1/2})$}
24: $C_{XX}^{-1/2} \leftarrow V_{XX} \, \text{diag}\big(\lambda_{XX}^{-1/2}\big) V_{XX}^T$
25:
26: **Correlation Matrix**
27: $C_{\mathbb{W}} \leftarrow C_{Y_+Y_+}^{-1/2} \ C_{Y_+X} \ C_{XX}^{-1/2}$   {Calculate the standardized cross-correlation matrix.}
28:
29: **Singular Value Decomposition (SVD)**
30: $(U, S, V) \leftarrow \text{SVD}(C_{\mathbb{W}})$   {Perform SVD to compute the CCA singular values.}
31: $(\rho_1, \rho_2, \ldots, \rho_r) \leftarrow diag(S)$   {$r$ is the embedding dimension.}
32: CCA-bound $\leftarrow \sum_{i=1}^{r}(1 - \rho_i^2)$   {Calculate CCA Bound based on Theorem-(3.2).}
33:
34: **Weight Matrix and Bias**
35: $W \leftarrow C_{YX} C_{XX}^{-1}$   {NBL weight.}
36: $b \leftarrow E[Y] - W E[X]$   {NBL bias.}

---

Algorithm 2 provides a detailed procedure for NBL parameter estimation and CCA-bound computation. This approach approximates self-attention layers by analytically deriving linear weights and biases while quantifying redundancy through Canonical Correlation Analysis (CCA). It takes the input $X$ and output $Y$ of any given self-attention layer. To capture the full behavior of the outputs, the residual connection is added as $Y_+ = Y + X$. While the CCA-bound is calculated based on the output of the residual connection, the linearization weights and biases are computed directly from the attention output. As a result, the residual connection is retained in the compressed layer. Although it is possible to calculate the weights directly from the residual connection output and remove the residual connection in the compressed model, its inclusion adds minimal complexity to the model and helps preserve the input information more effectively.

First, the algorithm calculates $E[X]$, $E[Y]$ and $E[Y_+]$, then unbiased estimates of cross-covariances matrices, $C_{YX}$ and $C_{Y_+X}$, and also auto-covariances, $C_{XX}$ and $C_{Y_+Y_+}$. Further, based on these covariance matrices, eigen-decomposition is applied, which gives eigenvalues and eigenvectors for

calculating inverse square roots of covariance matrices $C_{Y_+Y_+}^{-1/2}$ and $C_{XX}^{-1/2}$. The CCA standardized cross-correlation matrix ($C_{\mathbb{W}}$) is then computed using the cross-covariance matrix $C_{Y_+X}$ and with the inverse square roots of both input and output covariance matrices. Thereafter, $\rho_i$, the CCA singular values from SVD over $C_{\mathbb{W}}$ are computed. The CCA-bound quantifies the redundancy present in the attention layer by computing the total of $1 - \rho_i^2$ summed over all dimensions of the embeddings.

At last, the weight matrix and bias of the NBL are computed. The weight matrix is computed from the cross-covariance ($C_{YX}$) and inverse ($C_{XX}$), whereas the bias can be induced from the expression ($b = E[Y] - WE[X]$). This method realizes not only a linear approximation of the attention layer but also a rigorous quantification of the redundancy, hence this turns out to be a strong approach toward the compression and optimization of layers.

### D.1   Computational cost of the calibration

The proposed algorithm for NBL parameter estimation and CCA-bound computation is designed to be computationally efficient, making it feasible for large-scale models such as `Mistral-7B`, `Llama-3.1-8B` and `Llama-3.1-8B`. The primary computational steps involve mean and covariance computation, symmetric matrix eigen-decomposition, and singular value decomposition (SVD). These operations dominate the overall runtime complexity, with eigen-decomposition and SVD scaling as $O(d^3)$, where $d$ is the embedding dimension of the attention layers.

The algorithm first computes the mean of input and output activations, requiring $O(s \cdot t \cdot d)$ operations, where $s$ is the number of sequences and $t$ is the context length. Covariance matrices, including cross-covariances, are estimated with a complexity of $O(s \cdot t \cdot d^2)$. The eigen-decomposition of covariance matrices and the SVD of the standardized correlation matrix both scale as $O(d^3)$. Matrix inversion for computing the weight matrix also has a complexity of $O(d^3)$, while bias computation is relatively inexpensive at $O(d^2)$.

Overall, the total complexity of the algorithm is $O(d^3 + s \cdot t \cdot d^2)$. The method is efficient due to optimized linear algebra routines and direct computations. Additionally, the algorithm requires no hyperparameter tuning or fine-tuning beyond specifying the number of calibration samples.

### D.2   Calibration runtime

The main computational overhead in the calibration process arises from Algorithm 2, which performs Canonical Correlation Analysis (CCA) and computes the associated linear parameters. To empirically assess the runtime, we run the algorithm using `float32` precision on randomly initialized matrices that simulate the attention representations of different model scales in the Llama-3.1 family, as presented in Tables 1 and 7. Each experiment used 256 calibration samples with a context length of 2048. Since the calibration is applied independently to each attention layer, the total runtime for a model can be estimated by multiplying the per-layer runtime by the number of layers.

For the GPU-based implementation, the tensors required for the `Llama-3.1-405B` scale—with the specified precision, sample size, and context length—do not fit within the memory of a single NVIDIA A100 (80 GB). Therefore, we report GPU results in Tables 1 and 7 using two A100 GPUs running in parallel across all three model scales, whereas all other runtimes in this paper are calculated with a single GPU. As shown in these analyses, although calibration can be computationally demanding at large scales on CPU, it remains tractable with GPU acceleration and becomes more efficient when the cost is amortized across layers and repeated use cases.

In our implementation, compressing the `Llama-3.1-8B` and `Mistral-7B` models—each containing 32 attention layers takes under 30 minutes using an NVIDIA A100 GPU (80 GB) for model inference and activation extraction, computing CCA bounds and NBL weights and biases. This runtime includes all necessary steps, including covariance estimation, SVD, and linear parameter computation. These

Table 7: Calibration Runtime for Different Model Scales (256 samples, 2048 context length)

| Model | Llama-8B | Llama-70B | Llama-405B |
|---|---|---|---|
| **Hidden size** ($d$) | 4092 | 8192 | 16384 |
| **Hidden layers** | 32 | 80 | 126 |
| **Runtime per layer (CPU)** | 62.16 sec | 324.79 sec | 1839.64 sec |
| **Total (CPU)** | $\sim$0.55 hr | $\sim$7.22 hr | $\sim$64.38 hr |
| **Runtime per layer (GPU)** | 26.04 sec | 78.91 sec | 371.79 sec |
| **Total (GPU)** | $\sim$0.24 hr | $\sim$1.75 hr | $\sim$13.01 hr |

results demonstrate the practicality of our method for compressing large-scale models with minimal overhead. Moreover, the implementation can be further accelerated through additional optimization and system-level improvements.

### D.3 Implementation details

The algorithm is implemented using `PyTorch` [Paszke et al., 2019] and `SciPy` [Virtanen et al., 2020] for tensor operations, linear algebra routines, and eigen-decomposition, and `HuggingFace Transformers` [Wolf, 2019] for loading and managing the pretrained models. These libraries provide optimized and robust functionality for large-scale matrix operations, enabling efficient computation on standard hardware. Additionally, the implementation builds on the code repository provided by He et al. [2024] [1].

After passing the calibration data through the model and computing the activations, typically on the GPU, the computations in Algorithm 2 can be performed on either the CPU or GPU, depending on the available compute resources. We provide efficient implementations for both options. Running the algorithm on the CPU helps reduce GPU memory overhead and ensures broader hardware compatibility, while using the GPU offers faster runtime for large-scale models. `PyTorch` and `SciPy`'s linear algebra routines ensure efficient execution of the covariance, eigen-decomposition, and SVD steps efficiently on either backend, enabling smooth integration with diverse workflows.

## E Experiments

### E.1 Performance - efficiency 2D plots

We have included performance-efficiency plots with the pooled standard error (SE) intervals, where we visualize accuracy against both throughput and KV-cache gains, for the two best performing methods, Attention NBL and Attention DROP. The plots for `Llama-3.1-8B`, `Mistral-7B`, DeepSeek-Distill-Llama-8B and `Llama-3.1-8B` can be found in Figure 4. To evaluate the statistical significance of the differences, we used the pooled standard error (Pooled_SE) to aggregate the uncertainty across tasks when reporting average performance. The Pooled_SE is computed as:

$$\text{Pooled\_SE} = \frac{1}{n}\sqrt{\sum_{i=1}^{n} \text{SE}_i^2},$$

where $\text{SE}_i$ denotes the standard error of task $i$, and $n$ is the total number of tasks. We have n=8 benchmarks similar to the setting in Tables 2, 3, 4. The plots 4a, 4b, 4c indicate that, ***particularly at higher compression rates, NBL demonstrates a more favorable trade-off between KV-cache usage, accuracy, and throughput***, suggesting statistically significant Pareto optimality.

---

[1] https://github.com/CASE-Lab-UMD/LLM-Drop

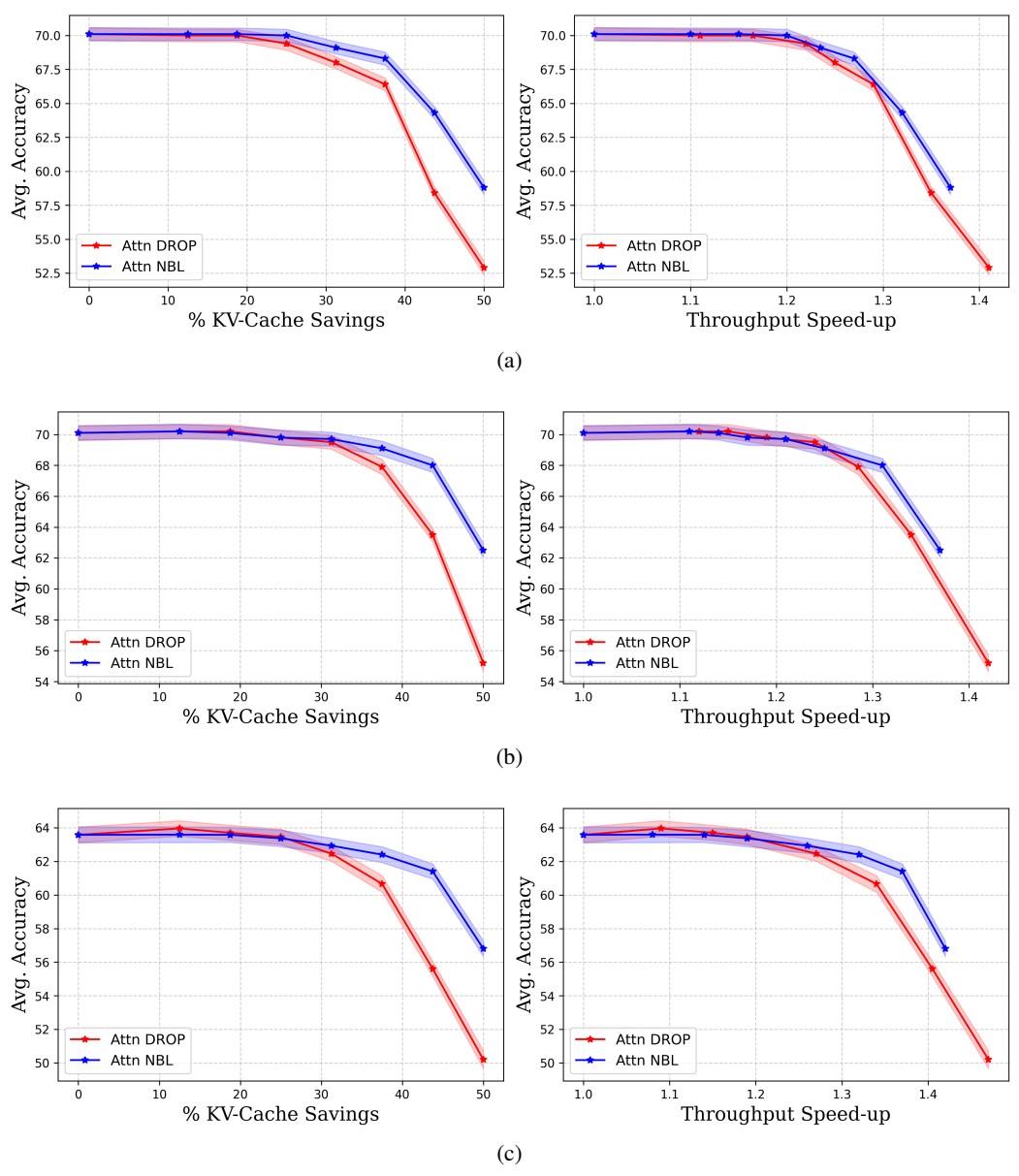

Figure 4: Accuracy versus (left) KV cache savings (right) throughput speed-up across varying compression levels for NBL and DROP, on (a) **Mistral-7B** (b) **Llama-3.1-8B** (c) **DeepSeek-R1-Distill-Llama-8B** models with the corresponding Standard Error intervals.

## E.2 Accuracy and speed-up comparisons with intervals

In this section, we present Tables 8, 9, 10, and 11 with the average accuracy, prefill and throughput speed-ups with standard error and standard deviation intervals for our Attention NBL method and the baseline Attention DROP, corresponding to the results in the main text.

Table 8: **Mistral-7B**: Accuracy and Speedup (Table 2).

| Method | Avg. Acc. ± SE (↑) | Prefill Speed-up (↑) | Throughput Speed-up (↑) |
|---|---|---|---|
| Base | 70.1 ± 0.41 | 1 | 1 |
| Drop-4 | 70.0 ± 0.41 | 1.08 ± 0.03 | 1.10 ± 0.02 |
| Drop-8 | 69.4 ± 0.41 | 1.18 ± 0.05 | 1.22 ± 0.01 |
| Drop-12 | 66.5 ± 0.42 | 1.29 ± 0.04 | 1.29 ± 0.03 |
| Drop-16 | 52.9 ± 0.42 | 1.44 ± 0.05 | 1.41 ± 0.01 |
| NBL-4 | 70.1 ± 0.41 | 1.08 ± 0.04 | 1.10 ± 0.01 |
| NBL-8 | 70.0 ± 0.41 | 1.17 ± 0.03 | 1.20 ± 0.02 |
| NBL-12 | 68.3 ± 0.42 | 1.28 ± 0.05 | 1.27 ± 0.01 |
| NBL-16 | 58.8 ± 0.42 | 1.40 ± 0.04 | 1.37 ± 0.02 |

Table 9: **Llama-3.1-8B**: Accuracy and Speedup (Table 3).

| Method | Avg. Acc. ± SE (↑) | Prefill Speed-up (↑) | Throughput Speed-up (↑) |
|---|---|---|---|
| Base | 70.1 ± 0.41 | 1 | 1 |
| Drop-4 | 70.2 ± 0.41 | 1.08 ± 0.035 | 1.11 ± 0.02 |
| Drop-8 | 69.8 ± 0.41 | 1.18 ± 0.048 | 1.19 ± 0.01 |
| Drop-12 | 67.9 ± 0.42 | 1.27 ± 0.033 | 1.29 ± 0.03 |
| Drop-16 | 55.2 ± 0.43 | 1.43 ± 0.025 | 1.42 ± 0.01 |
| NBL-4 | 70.2 ± 0.41 | 1.08 ± 0.040 | 1.11 ± 0.01 |
| NBL-8 | 69.8 ± 0.42 | 1.16 ± 0.031 | 1.17 ± 0.03 |
| NBL-12 | 69.1 ± 0.42 | 1.24 ± 0.027 | 1.25 ± 0.02 |
| NBL-16 | 62.5 ± 0.42 | 1.39 ± 0.043 | 1.37 ± 0.01 |

Table 10: **DeepSeek-R1-Distill-Llama-8B**: Accuracy and Speedup (Tables 4).

| Method | Avg. Acc. ± SE (↑) | Prefill Speed-up (↑) | Throughput Speed-up (↑) |
|---|---|---|---|
| Base | 63.6 ± 0.42 | 1 | 1 |
| Drop-4 | 63.9 ± 0.42 | 1.08 ± 0.04 | 1.09 ± 0.01 |
| Drop-8 | 63.5 ± 0.42 | 1.16 ± 0.04 | 1.19 ± 0.01 |
| Drop-12 | 60.7 ± 0.42 | 1.28 ± 0.05 | 1.34 ± 0.02 |
| Drop-16 | 50.2 ± 0.42 | 1.37 ± 0.04 | 1.47 ± 0.01 |
| NBL-4 | 63.6 ± 0.42 | 1.08 ± 0.04 | 1.08 ± 0.02 |
| NBL-8 | 63.4 ± 0.41 | 1.15 ± 0.04 | 1.19 ± 0.03 |
| NBL-12 | 62.4 ± 0.42 | 1.24 ± 0.05 | 1.32 ± 0.02 |
| NBL-16 | 56.8 ± 0.42 | 1.35 ± 0.05 | 1.42 ± 0.01 |

Table 11: **Llama-3.1-*70B*** (quant.): Accuracy and Speedup (Table 5).

| Method | Avg. Acc. ± SE (↑) | Prefill Speed-up (↑) | Throughput Speed-up (↑) |
|---|---|---|---|
| Base | 76.0 ± 0.40 | 1 | 1 |
| Drop-32 | 75.7 ± 0.40 | 1.20 ± 0.033 | 1.30 ± 0.01 |
| Drop-48 | 70.4 ± 0.40 | 1.29 ± 0.052 | 1.42 ± 0.02 |
| Drop-54 | 48.3 ± 0.42 | 1.34 ± 0.059 | 1.53 ± 0.02 |
| NBL-32 | 76.0 ± 0.41 | 1.16 ± 0.039 | 1.24 ± 0.03 |
| NBL-48 | 71.3 ± 0.41 | 1.25 ± 0.044 | 1.35 ± 0.01 |
| NBL-54 | 65.4 ± 0.43 | 1.30 ± 0.021 | 1.44 ± 0.02 |

### E.3 Qualitative analysis of LLM outputs after NBL

Here we present the output of `Llama-3-8B-Instruct`[2] model on a sample from the test set of GSM8K [Cobbe et al., 2021] as the input in Table 12. Here, the NBL gives the correct answer until 12 layer modification, whereas DROP fails to give the correct answer in similar compression.

Table 12: Correct answers are marked with the symbol "✓", and incorrect ones with "✗". All outputs are in response to a math word problem; the correct answer is extracted with the token "####".

| Question | Mark has a garden with flowers. He planted plants of three different colors in it. Ten of them are yellow, and there are 80% more of those in purple. There are only 25% as many green flowers as there are yellow and purple flowers. How many flowers does Mark have in his garden? |
|---|---|
| **Reference Answer** | There are 80/100 * 10 = «80/100*10=8»8 more purple flowers than yellow flowers. \n So in Mark's garden, there are 10 + 8 = «10+8=18»18 purple flowers. \n Purple and yellow flowers sum up to 10 + 18 = «10+18=28»28 flowers. \n That means in Mark's garden there are 25/100 * 28 = «25/100*28=7»7 green flowers. \n So in total Mark has 28 + 7 = «28+7=35»35 plants in his garden. \n #### 35 |

| Method | Output |
|---|---|
| **Baseline** | There are 10 yellow flowers. 80% more of those are purple. So 10 + (10 x 0.8) = 18 purple flowers. 10 + 18 = 28 yellow and purple flowers. 25% of that is 28 x 0.25 = 7 green flowers. 10 + 18 + 7 = 35 flowers. The answer is 35. \n #### 35 ✓ |
| **Attn NBL-8** | NBL-8 Ans: There are 10 yellow flowers. There are 10 * 1.8 = 18 purple flowers. There are 10 + 18 = 28 yellow and purple flowers. There are 25% as many green as yellow and purple, so there are 0.25 * 28 = 7 green. 7 + 28 = 35. 10 + 18 + 7 = 35. The answer is 35. \n #### 35 ✓ |
| **Attn NBL-12** | The number of yellow flowers is 10. The number of the other type of flower is 10 * 1.8 = 18. The number of green ones is 10 + 18 = 28, and 0.25 * 28 = 7. So the total number of flowers is 10 + 18 + 7 = 35. \n #### 35 ✓ |
| **Attn NBL-16** | The number of the first group (the one with the highest number) is 100% and the other two are 1.5 and 0.75. So the number of the first is 100% and the other two are 1.5 and 0.5. So the total number of the first is 100% and the other two are 1.5 and 1.5. ✗ |
| **Attn DROP-8** | There are 10 yellow flowers. There are 80% more of those in purple, so there are 10 + 0.8 * 10 = 18 purple flowers. The total number of yellow and purple flowers is 10 + 18 = 28. There are 25% as many green flowers as there are yellow and purple flowers, so there are 0.25 * 28 = 7 green flowers. The total number of flowers is 7 + 28 = 35. \n #### 35 ✓ |
| **Attn DROP-12** | There are 10 yellow flowers. 80% of 10 is 8, so there are 8 more, so 18 in total. 25% of 18 is 4, so there are 4 green. 10 + 8 + 4 = 22. So there are 22 of the non-green flowers. 4 is the green. So the total is 26. The answer is 26. ✗ |
| **Attn DROP-16** | We're given that the number of fish are is the main character, the most important thing is the of the in the problem. The problem states " 1, the " (the phrase "if and if, the " (the phrase "is a, the. The " (the " or " (a, but " (the. The is the " (the, the. (The " of the. The is the. . Let's analyze the first line, the. 1 1. is and, the. is and, the. is and, and. The is a, the. 1 and, the. is and, the. ✗ |

[2]https://huggingface.co/meta-llama/Meta-Llama-3-8B-Instruct

### E.4 Quantization of `Llama-3.1-70B` using the AWQ algorithm

The Activation-aware Weight Quantization (AWQ) algorithm Lin et al. [2024] is a hardware-friendly method designed for low-bit weight-only quantization of large language models (LLMs). AWQ focuses on preserving model performance while significantly reducing memory and computation costs. Below, we outline the mathematical framework and process to quantize the `Llama-3.1-8B`.

**Weight quantization:**

Quantization maps floating-point weights $\mathbf{W}$ to low-bit integers $\mathbf{Q}(\mathbf{W})$. For $N$-bit quantization, the function defined as:

$$\mathbf{Q}(\mathbf{W}) = \Delta \cdot \text{Round}\left(\frac{\mathbf{W}}{\Delta}\right), \quad \Delta = \frac{\max(|\mathbf{W}|)}{2^{N-1}},$$

where $\Delta$ is the quantization scale, and $\text{Round}(\cdot)$ maps values to the nearest integers. This reduces precision but can incur noticeable error if all weights are treated equally.

**Preserving salient weights:**

A key observation is that not all weights contribute equally to model performance. AWQ identifies the most important weights, termed *salient weights*, using activation magnitudes. Channels corresponding to large activation values are more likely to process critical features. Instead of mixed-precision quantization, AWQ applies a scaling factor $s > 1$ to these weights to reduce relative quantization error. For a weight $\mathbf{W}$, and input vectOR $\mathbf{X}$, the scaled quantization process is:

$$\mathbf{Q}(\mathbf{W} \cdot s) \cdot \frac{\mathbf{X}}{s} = \Delta' or \cdot \text{Round}\left(\frac{\mathbf{W} \cdot s}{\Delta'}\right) \cdot \frac{\mathbf{X}}{s},$$

where $\Delta'$ is the updated quantization scale. The ratio of quantization errors before and after scaling is given by:

$$\text{Error Ratio} = \frac{\Delta'}{\Delta} \cdot \frac{1}{s} \approx \frac{1}{s}, \quad \text{for } \Delta' \approx \Delta.$$

Thus, scaling effectively reduces the relative error for salient weights.

**Optimal scaling factor search:**

The AWQ algorithm optimizes the scaling factors $s$ by minimizing the output difference between quantized and original weights:

$$L(\mathbf{s}) = \left\| \mathbf{Q}\big(\mathbf{W} \cdot \text{diag}(\mathbf{s})\big)\big(\text{diag}(\mathbf{s})^{-1}\mathbf{X}\big) - \mathbf{W}\mathbf{X} \right\|.$$

where $\mathbf{s}$ is a per-channel scaling vector, and $\mathbf{X}$ represents cached activation features from a small calibration dataset. To simplify the search, AWQ uses the average activation magnitude per channel, $s_{\mathbf{X}}$, as a proxy for saliency:

$$\mathbf{s} = s_{\mathbf{X}}^{\alpha}, \quad \alpha^* = \arg\min_{\alpha} L(\mathbf{s}).$$

which allows an efficient search over a single exponent $\alpha$.

**AWQ and NBL:**

The AWQ algorithm's per-channel scaling and activation-aware strategy enable efficient compression without requiring retraining or large calibration datasets. AWQ is also designed to support layer fusion to optimize hardware efficiency. By fusing linear layers with adjacent operations, AWQ reduces intermediate memory access and kernel overhead, enhancing its performance even further. However, in the implementation used for the `Llama-3.1-70B` quantization in the NBL framework, we opted to use the unfused version for simplicity and flexibility during experimentation. Additionally, the linear weights calculated by NBL were also quantized by the AWQ method for compatibility. The speed-up measurements were done by generating 2048 tokens with a context length of 2048, and a batch size of 1, similar to the experiments with smaller models. We used the publicly available AutoAWQ [3] repository for the implementation of the AWQ on top of NBL.

Future work can address these limitations by integrating quantization-aware techniques directly into the NBL pipeline. For instance, a fully quantized NBL framework could adopt fine-grained scaling for linear layers while retaining the ability to fuse operations dynamically, thereby combining flexibility with optimal hardware utilization. Such advancements could further reduce memory overhead while maintaining model accuracy.

## E.5    Combining NBL with speculative decoding for faster inference

We evaluate the integration of speculative decoding and Neural Block Linearization (NBL) using the `DeepSeek-R1-Distill-Llama-8B` model [Guo et al., 2025]. Experiments are conducted on the MT-bench [Zheng et al., 2023] benchmark using an NVIDIA A100 GPU. For speculative decoding, we adopt the EAGLE-3 method [Li et al., 2025], which extends the traditional draft-and-verify framework by enabling the generation of multiple tokens per iteration with a low-cost draft model. In each decoding step, the draft model proposes a sequence of candidate tokens, which are then passed to a verifier model. The verifier checks these tokens in parallel, accepting the longest prefix that aligns with its own output. This process amortizes the cost of expensive autoregressive generation and significantly accelerates decoding.

To evaluate compatibility with NBL, we replace the standard verifier model in EAGLE-3 with its NBL-compressed counterpart. Since NBL preserves the input-output behavior of selected attention layers through linear approximations, the verifier remains functionally consistent while being more efficient. This substitution does not require changes to the speculative decoding logic or generation pipeline. For the draft model, we use the publicly available weights from the EAGLE repository [4].

Our results show that combining NBL with EAGLE-3 leads to compounding speed-ups, achieving up to 4.07× acceleration (see Table 6). This demonstrates that NBL is not only orthogonal to decoding-level acceleration strategies but can also enhance them with negligible integration overhead. These findings highlight NBL's practicality for real-world deployment scenarios, where both model size and inference latency are critical.

---

[3] `https://github.com/casper-hansen/AutoAWQ`
[4] `https://github.com/SafeAILab/EAGLE`

# F Ablation studies

## F.1 Dependency on the calibration dataset

Tables 13 and 14 present validation results on C4 and WikiText-2 for pruned or linearized **Llama-3.1-8B** and **Mistral-7B** models, comparing SLEB, Attn DROP, Attn NBL (each applied to 8 attention layers), and SliceGPT with 25% sparsity. These results are included to assess each method's dependency on the choice of calibration data used to prune or linearize LLMs.

Table 13: Perplexity Results on C4.

| Calibration ($\mathcal{D}$): | Wiki-2 | C4 | Wiki-2 | C4 |
|---|---|---|---|---|
| **Method** | Llama-3.1-8B | | Mistral-7B | |
| SliceGPT-25% | 123.29 | 25.41 | 28.28 | 13.07 |
| SLEB-8 | 20.53 | 20.54 | 15.07 | 15.49 |
| Attn DROP-8 | 11.40 | 11.35 | 9.20 | 9.20 |
| Attn NBL-8 | 12.22 | 11.37 | 9.85 | 9.08 |

Table 14: Perplexity Results on WikiText-2.

| Calibration ($\mathcal{D}$): | Wiki-2 | C4 | Wiki-2 | C4 |
|---|---|---|---|---|
| **Method** | Llama-3.1-8B | | Mistral-7B | |
| SliceGPT-25% | 14.27 | 74.33 | 7.61 | 12.11 |
| SLEB-8 | 16.70 | 16.67 | 10.23 | 11.13 |
| Attn DROP-8 | 7.39 | 7.48 | 6.81 | 6.81 |
| Attn NBL-8 | 7.18 | 9.57 | 5.59 | 7.33 |

## F.2 LoRA fine-tuning of NBL-linearized layers

We further investigate the potential of fine-tuning the linear layers derived by our NBL method using the Low-Rank Adaptation (LoRA) framework [Hu et al., 2022]. Specifically, we apply LoRA to the linearized layers on the same C4 dataset used for calculating the NBL parameters. This dataset closely resembles the original pretraining data, ensuring consistency in domain and distribution.

Our experiments use the `DeepSeek-R1-Distill-Llama-8B` model, where 12 and 16 attention layers are replaced with their NBL-linearized counterparts (denoted as Attn NBL-12 and Attn NBL-16). We then fine-tune these linearized layers using LoRA with a rank of 32, $\alpha = 64$, and a dropout of 0.1. Training is performed in `bfloat16` for 3 epochs with a learning rate of $1e-4$, an effective batch size of 16, and context length of 1024 tokens using a 5000-sample subset of the C4 validation split under a causal language modeling objective.

To further assess generalization beyond the calibration data, we also apply LoRA fine-tuning using the SlimPajama [Soboleva et al., 2023] dataset—a large-scale corpus with different statistical properties than C4. This setup allows us to explore LoRA's effectiveness under a domain-mismatched regime.

These results in Table 15 suggest that LoRA-based refinement offers only marginal improvements over NBL alone—e.g., 62.5% vs. 62.4% for NBL-12 and 58.2% vs. 56.8% for NBL-16, indicating that most of the performance gains stem from the underlying NBL mechanism itself. Furthermore, the consistent behavior across both matched and mismatched data regimes highlights the robustness and generality of NBL, where it provides competitive reasoning performance even without additional parameter-efficient fine-tuning, making it a lightweight and effective tool for compressing large pretrained language models.

Table 15: Average reasoning accuracy of NBL applied **DeepSeek-R1-Distill-Llama-8B** model before and after LoRA fine-tuning on C4 and SlimPajama pretraining datasets.

| Model Variant | Average Accuracy (%) |
|---|---|
| Baseline | 63.6 ± 0.42 |
| NBL-12 | 62.4 ± 0.42 |
| NBL-12 + LoRA (C4) | 62.5 ± 0.42 |
| NBL-12 + LoRA (SlimPajama) | 62.6 ± 0.42 |
| NBL-16 | 56.8 ± 0.42 |
| NBL-16 + LoRA (C4) | 58.2 ± 0.41 |
| NBL-16 + LoRA (SlimPajama) | 58.1 ± 0.43 |

### F.3 Attention heavy experiments with NVIDIA RULER benchmark

We report additional results on the RULER benchmark [Hsieh et al., 2024], which is designed to evaluate models on attention-intensive tasks requiring long-context reasoning and dependency tracking. Using the `DeepSeek-R1-Distill-Llama-8B` model, the results in Table 16 show that Attn NBL retains over 95% of baseline accuracy even when 2-4 attention layers are linearized, while SLEB exhibits a much sharper degradation. These findings highlight that NBL can selectively reduce computational cost in attention-heavy settings while preserving substantially better performance than recent baselines on long-context reasoning tasks.

Table 16: Average accuracy (%) of **DeepSeek-R1-Distill-Llama-8B** on NVIDIA RULER benchmark [Hsieh et al., 2024] at 4K context length).

| Method | Average Accuracy (%) |
|---|---|
| Baseline | 90.02 |
| SLEB-2 | 69.78 |
| SLEB-3 | 61.29 |
| Attn DROP-2 | 87.78 |
| Attn DROP-3 | 86.66 |
| Attn DROP-4 | 83.15 |
| Attn NBL-2 (ours) | 88.59 |
| Attn NBL-3 (ours) | 88.21 |
| Attn NBL-4 (ours) | 86.19 |

### F.4 CCA-Bound criterion vs cosine distance criterion

We analyze the performance of NBL under two different layer selection criteria: the CCA-bound criterion introduced in Theorem 3.2, and the cosine distance criterion originally employed in the DROP method of He et al. [2024]. Results for the cosine distance criterion are reported in Tables 17 and 18, with the last column ("CCA Avg.") showing the averages obtained from the CCA-bound criterion (Tables 2 and 3).

For the `Mistral-7B` model (Table 17), the cosine-based criterion performs comparably at smaller layer intervals (e.g., Attn NBL-4 and Attn NBL-8), but its performance degrades sharply for larger intervals (e.g., Attn NBL-12, Attn NBL-16), where both the average accuracy and CCA scores decline. Similar trends are observed for the `Llama-3.1-8B` model (Table 18), where performance drops from 70.2 for Attn NBL-4 to 58.0 for Attn NBL-16. These results suggest that the cosine distance criterion is less stable when approximating transformations across broader sections of the network, whereas the CCA-bound criterion provides greater reliability.

While the performance gap between CCA and cosine distance may appear less pronounced in some benchmarks, our choice of CCA is motivated by its stronger theoretical grounding. CCA explicitly measures the shared subspace between input and output representations, capturing structural alignments that preserve information flow. In contrast, cosine similarity only compares Euclidean angles, potentially overlooking linear dependencies. To illustrate this, consider a two-dimensional example with input and output tensors each consisting of three samples:

$$X = \begin{bmatrix} 1 & 0 \\ 0 & 1 \\ -1 & 0 \end{bmatrix}, \quad Y = \begin{bmatrix} 0 & 1 \\ -1 & 0 \\ 0 & -1 \end{bmatrix}.$$

There exists a transformation matrix $A$ where:

$$A = \begin{bmatrix} 0 & 1 \\ -1 & 0 \end{bmatrix}$$

such that $Y = XA$. Although this mapping is linear, the cosine similarity between each corresponding row of $X$ and $Y$ is zero, since the vectors are orthogonal. CCA, in contrast, identifies the full-rank

alignment of the shared subspace and assigns a maximal similarity score, reflecting the underlying linear connection. While this example is simplistic compared to the highly structured transformations in LLMs, it highlights a fundamental limitation of cosine-based criteria in detecting linearly aligned signal transformations.

These challenges become particularly relevant in attention-heavy tasks, where subtle but important features may be selectively amplified in sparse or nonlinear ways. Consequently, the difference between CCA and cosine distance is most pronounced in more demanding evaluations where errors in layer selection carry greater cost.

To validate this, we evaluated both criteria on the attention heavy NVIDIA RULER benchmark [Hsieh et al., 2024] under the same setting as in Table 16 using the DeepSeek-R1-Distill-Llama-8B and Llama-3.1-8B models in Tables 19 and 20 respectively, observing a consistent trend.

Across both models, the CCA-based criterion consistently outperforms the cosine-based criterion as the number of substituted layers increases, highlighting its advantage in guiding structure-aware substitutions and maintaining robustness in tasks that emphasize long-range reasoning. This empirical evidence further supports Theorem 3.2, which establishes the CCA-bound as a principled measure of the linear structure preserved between layers, and justifies its use as the preferred selection criterion for layer linearization with NBL.

Table 17: Performance of **Mistral-7B** with NBL using CCA vs cosine based criterion.

| Method | ARC-e (norm) | ARC-c (norm) | BoolQ | HellaSwag (norm) | MMLU (5-shot) | OBQA (norm) | PIQA (norm) | Wino-Grande | Cosine-Based Avg. (↑) | CCA-Based Avg. (↑) |
|---|---|---|---|---|---|---|---|---|---|---|
| Attn NBL-4 | 80.0 | 53.7 | 83.4 | 80.6 | 62.4 | 44.6 | 81.9 | 74.0 | 70.1 ± 0.40 | 70.1 ± 0.41 |
| Attn NBL-8 | 80.4 | 52.8 | 83.6 | 79.8 | 62.1 | 44.0 | 81.8 | 74.4 | 69.9 ± 0.40 | 70.0 ± 0.41 |
| Attn NBL-12 | 76.6 | 49.1 | 83.5 | 76.5 | 60.4 | 42.6 | 79.8 | 73.3 | 67.7 ± 0.42 | 68.3 ± 0.42 |
| Attn NBL-16 | 62.4 | 42.8 | 76.7 | 69.4 | 33.0 | 39.8 | 76.4 | 70.2 | 58.8 ± 0.40 | 58.8 ± 0.42 |

Table 18: Performance of **Llama-3.1-8B** with NBL using CCA vs cosine based criterion.

| Method | ARC-e (norm) | ARC-c (norm) | BoolQ | HellaSwag (norm) | MMLU (5-shot) | OBQA (norm) | PIQA (norm) | Wino-Grande | Cosine-Based Avg. (↑) | CCA-Based Avg. (↑) |
|---|---|---|---|---|---|---|---|---|---|---|
| Attn NBL-4 | 81.9 | 54.0 | 82.2 | 78.1 | 65.0 | 45.8 | 81.1 | 73.4 | 70.2 ± 0.43 | 70.2 ± 0.41 |
| Attn NBL-8 | 81.5 | 53.7 | 82.1 | 77.2 | 64.0 | 45.4 | 81.1 | 73.3 | 70.0 ± 0.41 | 69.8 ± 0.41 |
| Attn NBL-12 | 79.1 | 52.2 | 82.3 | 75.2 | 64.8 | 45.2 | 79.9 | 74.0 | 69.0 ± 0.41 | 69.1 ± 0.42 |
| Attn NBL-16 | 71.8 | 46.8 | 81.6 | 69.0 | 39.1 | 41.8 | 77.0 | 73.1 | 58.0 ± 0.42 | 62.5 ± 0.43 |

Table 19: Performance of **DeepSeek-R1-Distill-Llama-8B** with NBL using CCA vs cosine based criterion on NVIDIA RULER benchmark [Hsieh et al., 2024].

| Method Variant | Cosine-Based (%) | CCA-Based (%) |
|---|---|---|
| Baseline | 90.02 | 90.02 |
| Attn NBL-2 | 88.50 | 88.59 |
| Attn NBL-3 | 87.38 | 88.21 |
| Attn NBL-4 | 85.62 | 86.19 |

Table 20: Performance of **Llama-3.1-8B** with NBL using CCA vs cosine based criterion on NVIDIA RULER benchmark [Hsieh et al., 2024].

| Method Variant | Cosine-Based (%) | CCA-Based (%) |
|---|---|---|
| Baseline | 93.28 | 93.28 |
| Attn NBL-2 | 91.51 | 91.89 |
| Attn NBL-3 | 90.36 | 91.56 |
| Attn NBL-4 | 89.68 | 90.85 |

## F.5 Greedy selection

In this section, we perform an ablation with greedy selection, which iteratively checks for the differences in the bound scores and compresses the model incrementally in multiple iterations. Results are shown in Table 21 and 22, where the last column shows our NBL results from main body. Our results show that our NBL with CCA criterion outperforms greedy selection. Greedy linearization alters the activation distribution leading to inconsistencies in layer ranking. We also noticed that greedy selection chooses earlier layers, which are usually more less amenable to linearization, as shown in the main body.

Table 21: Performance of **Mistral-7B** with greedy selection.

| Method | ARC-e (norm) | ARC-c (norm) | BoolQ | HellaSwag (norm) | MMLU (5-shot) | OBQA (norm) | PIQA (norm) | Wino-Grande | Greedy Avg. (↑) | NBL Avg. (↑) |
|---|---|---|---|---|---|---|---|---|---|---|
| Greedy-4 | 79.9 | 53.4 | 83.8 | 80.6 | 62.4 | 44.2 | 81.9 | 73.9 | 70.0 ± 0.40 | 70.1 ± 0.41 |
| Greedy-8 | 79.3 | 51.7 | 82.7 | 79.1 | 62.2 | 43.6 | 81.0 | 73.6 | 69.2 ± 0.44 | 70.0 ± 0.41 |
| Greedy-12 | 69.2 | 46.1 | 81.0 | 74.1 | 46.3 | 40.2 | 79.2 | 72.9 | 63.6 ± 0.43 | 68.3 ± 0.42 |
| Greedy-16 | 60.6 | 40.8 | 76.1 | 68.5 | 34.3 | 36.8 | 67.8 | 70.2 | 57.7 ± 0.40 | 58.8 ± 0.42 |

Table 22: Performance of **Llama-3.1-8B** with greedy selection.

| Method | ARC-e (norm) | ARC-c (norm) | BoolQ | HellaSwag (norm) | MMLU (5-shot) | OBQA (norm) | PIQA (norm) | Wino-Grande | Greedy Avg. (↑) | NBL Avg. (↑) |
|---|---|---|---|---|---|---|---|---|---|---|
| Greedy-4 | 82.2 | 54.4 | 82.3 | 78.4 | 64.7 | 46.0 | 80.9 | 73.1 | 70.2 ± 0.42 | 70.2 ± 0.41 |
| Greedy-8 | 81.7 | 53.8 | 82.2 | 77.2 | 64.7 | 45.2 | 80.7 | 74.4 | 70.0 ± 0.43 | 69.8 ± 0.41 |
| Greedy-12 | 79.2 | 52.1 | 82.5 | 74.9 | 48.0 | 45.6 | 80.2 | 73.0 | 66.9 ± 0.40 | 69.1 ± 0.42 |
| Greedy-16 | 69.2 | 39.8 | 71.3 | 68.1 | 26.1 | 42.2 | 77.6 | 66.6 | 48.9 ± 0.41 | 62.5 ± 0.43 |

# G   Selected Transformer layers

Table 23 presents the sorted importance rankings of attention layers selected by the Attn DROP and Attn NBL methods across various models and calibration datasets. In all configurations, the methods consistently prioritize dropping or linearizing the higher-indexed layers, particularly those near the end of the model, suggesting that the later layers may be more redundant or compressible in terms of their contribution to overall performance. This pattern is prominent in all models, where the top-ranked layers for pruning are overwhelmingly concentrated toward the final half of the network.

Table 23: Sorted attention layer rankings selected by Attn DROP and Attn NBL across different models and calibration datasets. Lower ranks indicate higher importance.

| Method & Model | Sorted Layer IDs (Most to Least Important) |
|---|---|
| Attn DROP – `Mistral-7B` (C4) | 25, 26, 27, 24, 22, 28, 23, 30, 31, 21, 29, 20, 19, 18, 17, 16 6, 8, 9, 12, 14, 13, 11, 15, 4, 10, 7, 5, 3, 0, 2, 1 |
| Attn DROP – `Llama-3.1-8B` (C4) | 24, 25, 22, 23, 27, 26, 20, 28, 19, 29, 21, 18, 30, 17, 16, 15 31, 11, 10, 14, 13, 12, 8, 5, 6, 9, 4, 7, 3, 2, 1, 0 |
| Attn DROP – `DS-Distill-Llama-8B` (C4) | 24, 23, 22, 25, 26, 20, 27, 19, 28, 18, 21, 29, 17, 30, 16, 15 31, 11, 14, 10, 5, 13, 4, 9, 2, 8, 6, 3, 12, 7, 1, 0 |
| Attn DROP – `Llama-3-8B-Instruct` (C4) | 25, 24, 22, 23, 26, 28, 27, 20, 19, 21, 29, 18, 30, 17, 16, 31 15, 11, 14, 10, 13, 8, 12, 5, 9, 6, 4, 2, 3, 7, 1, 0 |
| Attn DROP – `Llama-3.1-70B` (quant.) (C4) | 62, 65, 59, 61, 63, 46, 50, 58, 48, 51, 53, 54, 57, 60, 64, 66 67, 68, 49, 69, 55, 56, 47, 44, 52, 42, 45, 70, 43, 40, 71, 41 72, 77, 79, 73, 76, 74, 78, 39, 38, 75, 36, 6, 25, 32, 7, 11 22, 37, 23, 26, 3, 24, 5, 30, 28, 29, 35, 33, 21, 4, 20, 10 2, 34, 31, 27, 8, 12, 9, 14, 18, 15, 1, 16, 19, 17, 13, 0 |
| Attn DROP – `Llama-3.1-8B` (WikiText-2) | 23, 24, 22, 25, 20, 19, 28, 26, 27, 21, 18, 29, 30, 17, 16, 15 31, 11, 0, 14, 13, 12, 9, 5, 8, 6, 4, 7, 3, 2, 1, 0 |
| Attn DROP – `Mistral-7B` (WikiText-2) | 22, 27, 23, 24, 25, 21, 26, 28, 20, 29, 30, 19, 18, 17, 16, 31 14, 15, 13, 9, 12, 8, 6, 10, 5, 11, 4, 7, 3, 2, 1, 0 |
| Attn NBL – `Mistral-7B` (C4) | 25, 26, 27, 24, 23, 22, 28, 30, 31, 29, 21, 20, 19, 18, 17, 16 6, 8, 9, 12, 14, 13, 11, 15, 4, 10, 7, 5, 3, 0, 2, 1 |
| Attn NBL – `Llama-3.1-8B` (C4) | 25, 26, 29, 28, 27, 24, 31, 23, 22, 20, 21, 19, 30, 18, 17, 16 11, 10, 15, 5, 14, 4, 12, 6, 9, 8, 13, 7, 2, 3, 0, 1 |
| Attn NBL – `DS-Distill-Llama-8B` (C4) | 25, 26, 29, 27, 24, 20, 23, 28, 22, 19, 21, 31, 30, 18, 17, 16 15, 11, 14, 10, 4, 5, 13, 9, 12, 6, 8, 7, 2, 3, 0, 1 |
| Attn NBL – `Llama-3-8B-Instruct` (C4) | 25, 24, 22, 23, 26, 28, 27, 20, 19, 21, 29, 18, 30, 17, 16, 31 15, 11, 14, 10, 13, 8, 12, 5, 9, 6, 4, 2, 3, 7, 1, 0 |
| Attn NBL – `Llama-3.1-70B` (quant.) (C4) | 62, 65, 59, 66, 63, 61, 46, 58, 55, 54, 50, 48, 53, 51, 57, 60 67, 69, 64, 68, 49, 47, 79, 70, 56, 52, 45, 72, 43, 44, 71, 42 76, 73, 74, 40, 78, 77, 41, 75, 38, 39, 26, 25, 24, 32, 36, 7 12, 28, 11, 4, 37, 30, 22, 23, 29, 9, 8, 20, 6, 27, 10, 33 34, 35, 21, 14, 31, 16, 15, 5, 3, 13, 19, 18, 17, 0, 2, 1 |
| Attn NBL – `Llama-3.1-8B` (WikiText-2) | 25, 26, 23, 24, 29, 27, 28, 20, 22, 31, 21, 19, 30, 18, 17, 16 11, 4, 5, 15, 10, 14, 12, 2, 9, 6, 13, 8, 3, 7, 0, 1 |
| Attn NBL – `Mistral-7B` (WikiText-2) | 27, 25, 26, 24, 22, 23, 28, 31, 30, 21, 29, 20, 19, 18, 17, 16 6, 8, 14, 4, 9, 13, 12, 15, 11, 10, 7, 5, 0, 2, 3, 1 |

# H   Inference complexity analysis and KV-cache usage details

This section provides a detailed explanation of the relationship between context length, prefill speed, and KV-cache usage in transformer models with NBL-applied attention layers.

## H.1   Figure analysis: prefill speed-up vs. context length

Figure 3 illustrates how prefill speed-up improves as more attention layers are modified by NBL. The baseline, represented by the black line, is normalized to a value of 1 across all context lengths. Each other line corresponds to different NBL configurations, where $m$ attention layers have been replaced with linear layers. The speed-up becomes more pronounced at longer context lengths, as the quadratic complexity $\mathcal{O}(n^2)$ of attention begins to dominate, while the linear layers operate at a lower complexity of $\mathcal{O}(nd)$. At shorter contexts, the differences between configurations are minimal since the computational overhead of attention is less significant. However, as the context grows, models with more linear layers (e.g., NBL-16) maintain higher prefill speeds due to the reduced cost of quadratic operations. This behavior aligns with the theoretical complexity expression:

$$\mathcal{O}((K - m) \cdot n^2 d + m \cdot nd)$$

where $K$ is the total number of attention layers, $m$ is the number of self-attention layers replaced by linear approximations through NBL, $n$ is the context length, and $d$ is the model embedding dimension. In this particular experiment setting to generate the figure, we used 2 NVIDIA A100 (80GB) GPU's, and a batch size of 16.

## H.2   KV-cache calculation for Grouped Query Attention based models

The KV-cache stores key and value tensors for each token processed by attention layers, enabling efficient incremental decoding. In models like Llama and Mistral, which use grouped-query attention, the KV-cache requirements depend on both the number of active attention layers and the context length. Grouped-query attention is designed to improve the efficiency of multi-head attention by reducing the redundancy in key-value storage across attention heads. In standard multi-head attention, each head independently maintains its own key and value tensors, leading to a cache size proportional to the total number of attention heads. Specifically, the cache size scales as $2 \cdot \text{Batch Size} \cdot n \cdot d$, where $n$ is the context length, $d$ is the hidden dimension of the model, and the factor of 2 accounts for both keys and values.

In grouped-query attention, multiple heads are organized into groups, with each group sharing a single key-value cache. Let $g$ denote the number of groups and $h$ the total number of attention heads and $bs$ denote the batch size. Instead of each head having separate key-value storage, all $h_g = h/g$ heads within a group use the same set of keys and values. As a result, the cache size is reduced by a factor of $g/h$, yielding the following expression for grouped-query cache size:

$$\text{KV-cache size} = 2 \cdot bs \cdot n \cdot d \cdot \frac{g}{h}$$

This optimization is particularly beneficial for large-scale models, where the number of attention heads can be substantial. By reducing redundant storage, grouped-query attention helps manage memory usage, especially for longer context lengths and large batch sizes. The application of NBL further optimizes the KV-cache by modifying a subset of the attention layers. When $m$ out of $K$ layers are replaced by linear layers, these layers no longer require key-value storage. Consequently, the KV cache size is reduced to:

$$\text{KV-cache size with NBL} = 2 \cdot bs \cdot n \cdot d \cdot \frac{g}{h} \cdot \frac{K - m}{K}$$

This dual optimization—grouped-query attention and NBL—substantially lowers the memory requirements during inference, allowing models like Llama and Mistral to handle longer sequences more efficiently. For example, the KV cache sizes in Table 24 demonstrate the impact of these optimizations. With a batch size of 64 and grouped-query attention (32 total heads divided into 8 groups), the cache size at a context length of 512 is 4 GB in the original configuration, reduced

to 2.5 GB with 12 layers modified by NBL. Similarly, at a context length of 4096, the cache size decreases from 32 GB to 20 GB. This reduction follows the expected scaling behavior, highlighting the effectiveness of NBL and grouped-query attention in optimizing memory usage.

The table values reflect the KV cache size scaling with context length. For instance, at a context length of 512, the original cache size is 4 GB, reduced to 2.5 GB with Attn NBL-12. At a context length of 4096, the cache size decreases from 32 GB to 20 GB. These reductions demonstrate how NBL reduces both computational complexity and cache requirements in large-scale transformers.

Table 24: KV-Cache sizes of both `Llama-3.1-8B` and `Mistral-7B` models for different context lengths with a batch size of 64 and half precision.

| Context Len. | Original (GB) | Attn NBL-4 (GB) | Attn NBL-8 (GB) | Attn NBL-12 (GB) | Attn NBL-16 (GB) |
|---|---|---|---|---|---|
| 512 | 4 | 3.5 | 3.0 | 2.5 | 2.0 |
| 1024 | 8 | 7.0 | 6.0 | 5.0 | 4.0 |
| 2048 | 16 | 14.0 | 12.0 | 10.0 | 8.0 |
| 4096 | 32 | 28.0 | 24.0 | 20.0 | 16.0 |
| 128000 | 1000 | 875.0 | 750.0 | 625.0 | 500.0 |

# I  Limitations and broader impacts

In this work, we present Neural Block Linearization (NBL), a novel and theoretically grounded approach for compressing LLMs without retraining. By identifying and substituting layers exhibiting high linear redundancy, NBL achieves significant inference speedups and memory savings across challenging reasoning benchmarks, including MMLU, HellaSwag, and ARC-Challenge. We derive the CCA-based bound to quantify input-output redundancy. One limitation is that certain nonlinear transformations in LLMs may not admit low-error linear approximations. This is an intrinsic property of deep architectures. Our framework explicitly accounts for such cases by ranking layers based on their approximation impact and applying substitutions only where low-error can be achieved. We further demonstrate that even local approximations result in strong end-to-end performance. The calibration data selection and local linearization considerations balance tractability and performance: we show empirically that NBL remains stable across diverse calibration sets and architectures.

**Broader impacts.**  Looking forward, NBL offers several promising directions for extension. Applying NBL to MLP sub-blocks, cross-attention layers, or even finer-grained structures like individual attention heads may unlock additional efficiency without sacrificing quality. Moreover, NBL is particularly well-suited for long-context inference, where its linearized attention yields increasing benefits in both speed and memory efficiency as sequence lengths grow. From a broader perspective, NBL contributes to the democratization and sustainability of AI by enabling faster, cheaper, and more private inference on edge devices and under-resourced environments. As compression becomes central to deploying foundation models at scale, techniques like NBL offer a reliable and interpretable path forward, supporting both practical utility and responsible innovation.

