# OpenReview forum: "Efficient Large Language Model Inference with Neural Block Linearization"
_NeurIPS.cc/2025/Conference — NeurIPS 2025 poster_

### Official Review · Reviewer_ZrVu · 2025-06-26

**Clarity:** 4
**Significance:** 2
**Originality:** 2
**Rating:** 4
**Confidence:** 3

**Summary:**

Neural Block Linearization (NBL) is a method that replaces blocks of a neural network with linear functions, specifically analytically derived Linear Minimum Mean Squared Error (LMMSE) estimators. The proposed selection criteria of layers for pruning is based on Canonical Correlation Analysis (CCA). NBL is best used to replace the attention layers in Transformer LLMs, and ablations are performed using it to replace entire Transformer blocks (attention + feedforward layers), speeding up the autoregressive generation process while maintaining reasonable performance compared to other structural pruning methods. The results demonstrate that certain attention layers in some models can be replaced with analytically computed linear functions with minimal decreases in performance and some improvement to generation speed.

**Questions:**

Can you report evaluations of SliceGPT and SLEB for Tables 4-5, as well as PruneNet (or other structured pruning baselines that you think are more suitable) for all performance evaluations?
Including more comparisons will make me more inclined to raise the score.

Can you report the level of sparsity for SLEB/Block/Attn Drop/NBL-m, similar to SliceGPT or other structured pruning methods?
This will make the sparsity comparison between methods more clear.

Can you add ablations on replacing the feedforward layers? Or, explain why the only ablations are on dropping attention layers or Transformer blocks (equivalent to dropping attention+feedforward layers)?

**Ethical Concerns:**

["NO or VERY MINOR ethics concerns only"]

**Final Justification:**

I feel that NBL may be interesting to the community, but am unconvinced by the argument for CCA over cosine distance.

RESOLVED:
* including additional baselines
* including explicit sparsity levels

UNRESOLVED:
* evidence showing superiority of CCA over cosine distance

**Limitations:**

yes

**Quality:**

2

**Strengths And Weaknesses:**

The submission is technically sound and clearly written.
The claims of increased inference speed with minimal loss of accuracy for the method are well supported within ablations on number and type of layers replaced.
However, the evaluation is limited as some structural pruning baselines are missing (such as PruneNet, ICLR 2025), and some models have no pruning baselines at all (DeepSeek-R1-Distill-Llama-8B and Llama-3.1-70B).
The levels of sparsity are unclear for the given methods in the tables besides SliceGPT.

Without fair comparison to pruning baselines, the significance of the results is diminished.

The results from the ablation on CCA-Bound criterion vs cosine distance, as well as for greedy selection are overclaimed: Tables 17-19 lack uncertainty estimates and do not show meaningful difference between CCA and the cosine/greedy criteria.

---

> ### Author Rebuttal · Authors · 2025-07-30
>
> We thank the reviewer for the feedback. Below we address the points raised.
>
> ---
> ## Strengths:
>
> > This paper is well-written and easy to follow.
>
> > NBL uses CCA to guide the transformation of attention layers, building on a good theoretical foundation.
>
> > Experimental results show the effectiveness of NBL on reasoning tasks.
>
> We appreciate the positive comments by the reviewer.
>
> ---
> ## Weaknesses:
> > W1. The evaluation is limited as some structural pruning baselines are missing.
>
> Thank you for the suggestion. We would like to clarify that Tables 2–5 already include evaluations against a strong structural pruning baseline, DROP (He et al. "What Matters in Transformers? Not All Attention is Needed", 2024) which is one of the most recent and competitive pruning methods. And for Mistral-7B and Llama-3.1-8B, also SliceGPT and SLEB. These represent some of the most recent and competitive attention-pruning techniques.
>
> Previously:
> - Tables 2 and 3 evaluate NBL against DROP, SliceGPT, and SLEB on Mistral-7B and Llama-3.1-8B, demonstrating NBL's favorable trade-off between accuracy and speed.
> -  Table 4 extends this to DeepSeek-R1-Distill-Llama-8B, a newly released model, again comparing with DROP.
> -  Table 5 explores scalability by applying NBL to a quantized Llama-3.1-70B model, where we aimed to show feasibility on even larger models.
>
> However, we appreciate this observation and agree that broader baseline coverage is useful as time permitted. In response to this feedback, we extended Table 4 to include PruneNET, SliceGPT, and SLEB, along with existing DROP and NBL entries:
>
> ### Extended Table-4: Deepseek-8B:
> |Method|Avg.Acc.(↑)|Prefill(↑)|Throughput(↑)|
> |-|-|-|-|
> |Baseline|63.6|1.00|1.00|
> |PruneNET-15%|53.7|1.08|1.08|
> |PruneNET-25%|46.4|1.16|1.15|
> |PruneNET-35%|40.8|1.22|1.19|
> |SliceGPT-15%|51.2|1.09|1.09|
> |SliceGPT-25%|46.3|1.15|1.13|
> |SliceGPT-35%|40.6|1.21|1.20|
> |SLEB-4(12.5%)|56.2|1.09|1.14|
> |SLEB-8(25%)|47.7|1.17|1.29|
> |SLEB-12(37.5%)|42.4|1.26|1.53|
> |Attn DROP-12|60.7|1.27|1.34|
> |Attn DROP-16|50.2|1.37|1.47|
> |Attn NBL-12|__62.4__|1.24|1.32|
> |Attn NBL-16|56.8|1.35|1.42|
>
> We note that PruneNET required parameter tuning for DeepSeek. The default learning rate (5e-4) for the action model training mentioned in manuscript led to poor convergence, and sweeping between 1e-5 and 1e-3, we found that 1e-4 yielded better results, but performing marginally better than SliceGPT.
>
> Through these results, we see that Attn NBL gives the least drop in accuracy, while significantly increasing the inference speed, suggesting a better trade-off compared to baselines. In the revised manuscript, we will also extend Tables 2 and 3 with the PruneNET evaluation for consistency.
>
> For Table 5, which evaluates Llama-3.1-70B quantized to 4-bit using AWQ, we could not include PruneNET or SliceGPT as **these methods do not currently support post-quantization pruning**. However, we have now included SLEB in addition to DROP and our method NBL:
>
> ### Extended Table-5: Llama-3.1-70B (quant.):
> |Method|Avg.Acc.(↑)|Prefill(↑)|Throughput(↑)|
> |-|-|-|-|
> |Baseline(quant.)|76.0|1.00|1.00|
> |SLEB-16|70.9|1.23|1.24|
> |SLEB-32|55.9|1.63|1.61|
> |Attn DROP-32|75.7|1.20|1.30|
> |Attn DROP-48|70.4|1.29|1.42|
> |Attn DROP-54|48.3|1.34|1.53|
> |Attn NBL-32|76.0|1.16|1.24|
> |Attn NBL-48|71.3|1.25|1.35|
> |Attn NBL-54|65.4|1.30|1.44|
>
> Again, both Attn NBL-32 and Attn NBL-48 outperform SLEB-16 in accuracy while matching or exceeding its inference throughput, with significantly lower KV-cache usage due to more attentions removed. These results further highlight that NBL consistently achieves a better accuracy/efficiency trade-off, even under aggressive compression settings.
>
> > W2. The levels of sparsity are unclear for the given methods in the tables besides SliceGPT.
>
> Thank you for this suggestion. While we agree that sparsity is an important comparative metric, our primary focus in this work is not to maximize sparsity, but rather to improve inference efficiency through the selective substitution of attention layers. Therefore, our main criteria while designing our method is to achieve a better "*accuracy / inference-speed*" or "*acuracy / KV-cache-reduction*" trade-offs.
>
> That said, we understand that some readers may be interested in comparing methods from the perspective of accuracy vs sparsity (**even though this is not the main objective of our paper**). In our main text, we already report the number of layers replaced or dropped, which corresponds directly to sparsity in block-based SLEB and Block DROP methods (e.g., dropping 8 blocks out of 32 gives 25% sparsity). For Attn NBL, we replace attention blocks with lightweight linear layers. These replacements preserve the interface but eliminate the computation of QKV projections and attention maps, significantly reducing FLOPs and runtime.
>
> In response to the reviewer’s comment, we have added explicit sparsity ratios for all methods in Table 2,3,4, along with their average reasoning accuracy, together with a wider discussion of the advantages and disadvantages of each method. Note that in all methods, the embedding and final projection layers remain intact, which comprise 1.81% and 6.54% of total parameters in Mistral-7B and Llama-3.1-8B, respectively. Therefore, we report sparsity only over the remaining layers.
>
> For Tables 2 (Mistral-7B) and 4 (DeepSeek-8B) the sparsity ratios are the same as Llama-3.1-8B, since the only differences lie in the size of the embedding and final projection layers.
>
> ### Table-3 extended with sparsities: Llama-3.1-8B:
> |**Method**|**Sparsity(%)**|**Avg. Accuracy(↑)**|
> |-|-|-|
> |**Baseline**|0|70.2|
> |SliceGPT-15%|15.0|58.8|
> |SliceGPT-25%|25.0|49.9|
> |SliceGPT-35%|35.0|43.4|
> |SLEB-4|12.5|59.7|
> |SLEB-8|25.0|46.5|
> |SLEB-12|37.5|42.5|
> |Block DROP-4|12.5|64.3|
> |Block DROP-8|25.0|40.2|
> |Block DROP-12|37.5|43.9|
> |**Block NBL-4**|11.5|67.2|
> |**Block NBL-8**|23.1|59.3|
> |**Block NBL-12**|34.6|50.4|
> |Attn DROP-4|2.4|__70.2__|
> |Attn DROP-8|4.8|__69.8__|
> |Attn DROP-12|7.2|67.9|
> |Attn DROP-16|9.6|55.2|
> |**Attn NBL-4**|1.5|__70.2__|
> |**Attn NBL-8**|2.9|__69.8__|
> |**Attn NBL-12**|4.3|__69.1__|
> |**Attn NBL-16**|5.8|62.5|
>
> As shown in table, operating only on attention layers naturally result in lower sparsity as the majority of parameters (~80%) reside in the MLP layers. However, attention layers are significantly more computationally expensive due to their $O(N^2)$ complexity, making them ideal targets for improving inference efficiency in practice.
>
> However, our proposed **Block NBL method achieves a notably strong accuracy/sparsity trade-off**. For instance, **Block NBL-12 yields 50.4 accuracy with 34.6% sparsity, whereas methods like Block DROP-8 gives 40.2%, SLEB-8 has 46.5% and SliceGPT-25% has 49.9% accuracies each with a lower 25% sparsity**. This highlights the effectiveness of our selective substitution strategy, particularly in maintaining accuracy
>
> Moreover, although inducing less sparsity by acting exclusively on attention layers, Attn NBL provides the best accuracy–throughput trade-off: achieving less than 1% accuracy drop while improving throughput by over 32% on DeepSeek, also suggesting pareto-optimality (Appendix-E.3). This demonstrates that even minimal modifications, when strategically applied to computational bottlenecks, can lead to substantial efficiency gains without sacrificing accuracy.
>
> > W3. The results from the ablation on CCA-Bound criterion vs cosine distance, as well as for greedy selection are ... no uncertainty estimates.
>
> We include the uncertainty estimations for the ablation with cosine distance criterion, as in Tables 17 and 18:
>
> Mistral-7B:
> |Method|Cosine Dist.Avg.Acc.±SE (%)|CCA Avg.Acc.±SE(%)|
> |-|-|-|
> |Attn NBL-4|70.1±0.40|70.1±0.41|
> |Attn NBL-8|69.9±0.40|70.0±0.41|
> |Attn NBL-12|67.7±0.42|68.3±0.42|
> |Attn NBL-16|58.8±0.40|58.8±0.42|
>
> Llama-3.1-8B:
> |Method|Cosine Dist.Avg.Acc.±SE (%)|CCA Avg.Acc.±SE(%)|
> |-|-|-|
> |Attn NBL-4|70.2±0.43|70.2±0.41|
> |Attn NBL-8|70.0±0.41|69.8±0.41|
> |Attn NBL-12|69.0±0.41|69.1±0.42|
> |Attn NBL-16|58.0±0.42|62.5±0.43|
>
> While the results are mostly similar, we observe statistically significant improvements in specific settings. For example, on Mistral-7B, Attn NBL-12 with the CCA criterion outperforms cosine distance by 1.5× the standard error. On LLaMA-3.1-8B, the gap is even more pronounced for Attn NBL-16, where CCA yields a 10× SE improvement.
>
> These results suggest that the CCA-based selection is more stable and effective, particularly in deeper linearization settings. Furthermore, although the accuracy results of NBL versus the greedy selection are similar, NBL is preferred as it is computationally more efficient by eliminating the iterative procedure.
>
> ---
> ## Questions:
>
> > Q1. Can you report evaluations of SliceGPT and SLEB for Tables 4-5, as well as PruneNet? More comparisons will make me more inclined to raise the score.
>
> Please refer Weakness-1, for this matter.
>
> > Q2. Can you report the level of sparsity for SLEB/Block/Attn Drop/NBL-m, similar to SliceGPT?
>
> Please refer Weakness-2 for this matter.
>
> > Ablations on replacing the feedforward layers? Or, explain why the only ablations are on dropping attention layers or Transformer blocks?
>
> We chose not to include ablations on dropping only the feedforward (MLP) layers because prior work, specifically the DROP paper has shown that modifying or removing MLP layers not only leads to a significant degradation in reasoning performance but also yields negligible improvements in inference speed, particularly in terms of KV-cache efficiency. In contrast, targeting the $O(N^2)$ attention layers—either individually (Attention-NBL) or jointly with MLPs via full block removal (Block-NBL)—has demonstrated clear and consistent benefits across both accuracy and speed. Therefore, we focused our experiments on these interventions that we believe to be more impactful.
>
> ---
> We hope our responses address the reviewer’s questions and are happy to clarify any remaining points.
>
> Best,

---

> > ### Comment · Reviewer_ZrVu · 2025-08-04
> >
> > Thanks for the additional baselines, which I believe better position the work. I have increased my score in response.
> >
> > Thank you for the response to "W3 The results from the ablation on CCA-Bound criterion vs cosine distance...", but I do not agree that the results you shared "suggest that the CCA-based selection is more stable and effective".
> > The results show no difference for 3/4 of the experimental settings with no clear trend.
> >
> > I feel that NBL may be interesting to the community, but am unconvinced by the argument for CCA over cosine distance.
> >
> > ---
> >
> > RESOLVED:
> > * including additional baselines
> > * including explicit sparsity levels
> >
> > UNRESOLVED:
> > * evidence showing superiority of CCA over cosine distance
> >
> > Therefore, I will recommend borderline accept unless a more compelling argument is presented.

---

> ### Author Response · Authors · 2025-08-07
> **Response to Reviewer ZrVu's Comments**
>
> Thank you once again for your thoughtful feedback and for increasing your score in response to the added baselines and sparsity reporting.
>
> While the performance advantage of the CCA-based criterion over the cosine-based might be less clear in some tasks, our motivation for adopting CCA is based on its stronger theoretical grounding. CCA captures how information is preserved or transformed across a layer by identifying the shared subspace between its input and output representations. In contrast, cosine similarity compares only the Euclidean angle between these representations, potentially missing structurally meaningful alignments.
>
> To illustrate this, consider the following $X,Y \in ℝ^{3×2}$ that is a 2 dimensional example of input and output tensors, each with 3 corresponding samples:
>
> $X$ = $\begin{bmatrix}
> 1 & 0 \newline
> 1 & 1 \newline
> 2 & -1
> \end{bmatrix}$, $\quad$ $Y$ = $\begin{bmatrix}
> 0 & -1 \newline
> 1 & -1 \newline
> -1 & -2
> \end{bmatrix}$.
>
> In this case, there exists a matrix $A ∈ ℝ^{2×2}$ such that $Y = X A$. Concretely we can construct the linear transformation as:
>
> $Y$ = $X$ $\cdot$ $\begin{bmatrix}
> 0 & -1 \newline
> 1 & 0
> \end{bmatrix}$,
>
> where:
>
> $A$ = $\begin{bmatrix}
> 0 & -1 \newline
> 1 & 0
> \end{bmatrix}$.
>
> However, the cosine similarity between each corresponding row $x_i$ and $y_i$ is zero, due to orthogonal directions. CCA, in contrast, correctly identifies the full-rank alignment of the shared subspace and assigns a similarity score of the highest due to linear connection where each canonical correlation coefficient $\rho_i = 1$.
>
> **While this is a simple example illustrating a failure case of cosine similarity, we acknowledge that the transformations learned by LLMs are far more complex and structured than in this simple example.** Nonetheless, the example highlights a fundamental limitation of cosine-based criteria in detecting close-to-linearly-aligned signal transformations.
>
> These challenges are particularly relevant in more complex attention-heavy tasks, where certain attention heads selectively amplify subtle but important features, often in a sparse or highly nonlinear manner. Therefore, the difference between the CCA and cosine based criteria may become clearer in more difficult tasks such as the NVIDIA RULER benchmark (same as in Q2 of Reviewer 8B3g), where the cost of incorrect layer selection is higher, whereas easier benchmarks may exhibit noisier behavior.
>
> **To validate this in more realistic scenarios, we evaluated both criteria on the challenging NVIDIA RULER benchmark, which emphasizes long-range dependencies and places greater weight on attention mechanisms.** The results on the DeepSeek-R1-Distill-Llama-8B model are shown below:
>
> **DeepSeek-R1-Distill-Llama-8B:**
>
> | Method Variant | CCA-Based (%) | Cosine-Based (%) |
> |----------------|---------------|------------------|
> | Baseline       | 90.02         | 90.02            |
> | Attn NBL-2     | **88.59**     | 88.50            |
> | Attn NBL-3     | **88.21**     | 87.38            |
> | Attn NBL-4     | **86.19**     | 85.62            |
>
> We have also replicated the same setting using the Llama-3.1-8B model and observed a consistent pattern:
>
> **Llama-3.1-8B:**
>
> | Method Variant | CCA-Based (%) | Cosine-Based (%) |
> |----------------|---------------|------------------|
> | Baseline       | 93.28         | 93.28            |
> | Attn NBL-2     | **91.89**     | 91.51            |
> | Attn NBL-3     | **91.56**     | 90.36            |
> | Attn NBL-4     | **90.85**     | 89.68            |
>
> Across both models and increasing compression settings, the CCA-based selection criterion yields consistently better performance than cosine-based, highlighting its advantage in guiding structure-aware layer substitutions.
>
> We will do our best to incorporate these new results and clarify the underlying motivation in the revision. Thank you again for your constructive and thoughtful feedback, which helped improve both the clarity and empirical depth of our work.

---

> > ### Comment · Reviewer_ZrVu · 2025-08-07
> >
> > As you state, the internal transformations learned by layers are probably not as simple as the given example.
> >
> > I am surprised that the Llama-3.1-8B baseline is 5 points higher than the number (88.3) in the RULER repository, and see that the improvement in accuracy of using CCA over Cosine ranges from 0.0009 to 0.0120, replacing only 2-4 attention layers instead of 4-16.
> >
> > The theoretical grounding is believable, but stronger empirical results should validate the choice.

---

> > > ### Author Response · Authors · 2025-08-09
> > > **Response to Reviewer ZrVu**
> > >
> > > We thank the reviewer for the thoughtful follow-up and continued engagement.
> > >
> > > We run all experiments via the widely-used *Eleuther AI*'s `lm-evaluation-harness` library to ensure compatibility with our existing codebase, rather than with the NVIDIA repository, which may introduce variations in formatting, where the relative gap between selection criteria, *i.e., from the same baseline*, suggests cosine similarity might select less suitable layers, while we also agree it is interesting to expand the evaluation.
> > >
> > > We appreciate the reviewer’s recognition of the theoretical grounding behind the use of CCA. We will clarify these points in the revision. Thank you again for your feedback.
> > >
> > > Best regards,

---

### Official Review · Reviewer_8B3g · 2025-06-29

**Clarity:** 3
**Significance:** 3
**Originality:** 3
**Rating:** 4
**Confidence:** 4

**Summary:**

This paper introduces Neural Block Linearization (NBL), transforming the self-attention layers into linear approximations by Linear Minimum Mean Squared Error estimators. After transforming the attention layers from the existing LLM into linear layers, NBL effectively balances inference speed-up with accuracy.

**Questions:**

1. The capability of linear layers may be limited by the context length of the calibration set. Can using a calibration set with relatively short context length still be good at handling long context?
2. For tasks that require long-distance dependencies, will this linearization cause a "cliff-like" drop in model performance?

**Ethical Concerns:**

["NO or VERY MINOR ethics concerns only"]

**Limitations:**

yes

**Quality:**

3

**Strengths And Weaknesses:**

Strengths:
1. This paper is well-written and easy to follow.
2. NBL uses the Canonical Correlation Analysis (CCA) to guide the transformation of attention layers, building on a good theoretical foundation.
3. Experimental results show the effectiveness of NBL on reasoning tasks.

Weaknesses:
1. Computing the linear approximations requires a longer time than other pruning methods (e.g., SliceGPT).
2. Some typos:
- Inconsistent method name between “Attn SLEB-m” in Line 218 and ‘’SLEB-m’’ in Table 2, 3.
- As the shape of $X$ is $\mathbb{R}^{(s \cdot t)} \times h_{in}$ in equation(1), equation (2) should be $\hat{Y} = X W + b$.

---

> ### Author Rebuttal · Authors · 2025-07-30
>
> We thank the reviewer for the feedback. Below we address the points raised.
>
> ---
> ## Strengths:
>
> > This paper is well-written and easy to follow.
>
> > NBL uses the Canonical Correlation Analysis (CCA) to guide the transformation of attention layers, building on a good theoretical foundation.
>
> > Experimental results show the effectiveness of NBL on reasoning tasks.
>
> We appreciate the positive comments by the reviewer.
>
> ---
> ## Weaknesses:
>
> > W1. Computing the linear approximations requires a longer time than other pruning methods (e.g., SliceGPT).
>
> Thank you for raising this point. It is true that our calibration step is somewhat more computationally intensive than simpler methods like DROP, primarily because it involves computing second-order statistics (covariance matrices) and performing SVD. However, our method takes comparable time compared to SliceGPT. As shown in our Table 1, pruning an 8B Llama-3 model takes under 30 minutes on an A100 GPU. In comparison, SliceGPT reports a runtime of 44 minutes to compress a Llama-2 7B model on a more powerful H100 GPU (Table 3 of their paper).
>
> However, we emphasize that this step is performed **only once, offline**, and **does not require gradient-based optimization or fine-tuning**. As shown in Table 1, the wall-clock time remains practical even for 405B sized models (e.g., under 30 minutes for 8B models, in around 13 hours for 405B models).
>
> Importantly, our method yields more robust accuracy retention across tasks (see Tables 2–5), particularly in the mid-to-high compression regimes where pruning-based approaches like SliceGPT degrade more rapidly. We believe this slight increase in calibration cost is justified by the **better trade-off between efficiency and accuracy**, and by the fact that NBL can be applied plug-and-play to any pre-trained model without retraining.
>
> ---
> > W2. Some typos:
> > - Inconsistent method name between “Attn SLEB-m” in Line 218 and “SLEB-m” in Table 2, 3.
> > - As the shape of $X$ is $\mathbb{R}^{(s \cdot t)} \times h_{\text{in}}$ in Equation (1), Equation (2) should be $\hat{Y} = XW + b$.
>
> Thank you for pointing out these. There was a typo in Line 218 as SLEB acts on te whole transformer block, therefore all the methods that refers to SLEB are the same and will be changed to “SLEB-m”. Furthermore, we will revise the definition of $X$ and $Y$ in Line 127 to be the transpose of the current version, and become $h_{\text{in}} \times \mathbb{R}^{(s \cdot t)}$ and $h_{\text{out}} \times \mathbb{R}^{(s \cdot t)}$ respectively. This way, the Equation (2) will correctly match the matrix dimensions, and be consistent with the rest of the paper.
>
> ---
> ## Questions:
>
> > Q1. The capability of linear layers may be limited by the context length of the calibration set. Can using a calibration set with relatively short context length still be good at handling long context?
>
> Thank you for the insightful question. It's important to emphasize that the purpose of the calibration set in NBL is not to emulate full-context inference, but to assess the **local linearity and redundancy structure** of each attention layer. Since our method substitutes layers based on the canonical correlation between inputs and outputs, rather than their absolute sequence-level behavior, very long contexts were not required during calibration.
>
> In practice, we use sequences of 2048 tokens, which already cover a wide range of attention behaviors for linear substitution. The selected calibration length aligns with the default context range for the evaluated models (e.g., Mistral, LLaMA-3.1, DeepSeek), and suffices to capture the essential token-wise interaction patterns for reasoning tasks.
>
> ---
> > Q2. For tasks that require long-distance dependencies, will this linearization cause a "cliff-like" drop in model performance?
>
> Thank you for the question. In our evaluations, we did not observe a cliff-like collapse in performance. Instead, the degradation is generally gradual and varies by task complexity. Some benchmarks, particularly those relying on shallow reasoning or factual recall (e.g., BoolQ, HellaSwag), are more resilient to linearization. Others, such as ARC-e or multi-hop MMLU categories, can be more sensitive depending on which layers are replaced. This variability reinforces the value of our CCA-based selection criterion, which helps identify layers whose output structure is already close to linear and less critical for modeling long-range dependencies.
>
> That said, as context length increases, **the computational cost of attention dominates**, and even **replacing a small number of attention layers** yields substantial gains. This is evidenced by Figure 3, which shows growing speedups at longer contexts even when only a few layers are substituted, and Table-21 that suggests huge KV-cache savings in longer contexts. Thus, for tasks that require long-range reasoning, **a conservative application of NBL still provides meaningful efficiency improvements without sacrificing core performance**.
>
> To evaluate robustness under such conditions, we report results on the NVIDIA RULER benchmark (Hsieh et al., 2024), which consists of attention-heavy tasks requiring reasoning, using DeepSeek-R1-Distill-Llama-8B model. RULER is specifically designed to test a model’s ability to track and integrate long-range dependencies, making it a suitable stress test for compression methods like NBL. Here our average accuracies over the RULER benchmarks:
>
> | Method   | Avg. Accuracy (%) |
> | --------------- | ------------ |
> | **Uncompressed Baseline**  | 90.02        |
> | **SLEB-2**      | 69.78        |
> | **SLEB-3**      | 61.29        |
> | **Attn DROP-2** | 87.78        |
> | **Attn DROP-3**| 86.66        |
> | **Attn DROP-4** | 83.15        |
> | **Attn NBL-2 (ours)**  | **88.59**    |
> | **Attn NBL-3 (ours)**  | **88.21**    |
> | **Attn NBL-4 (ours)**  | **86.19**    |
>
> These results show that Attn NBL retains over 95% of full accuracy even with 2–4 attention layers linearized, whereas SLEB degrades much more rapidly. This demonstrates that NBL can selectively reduce computational cost in attention-heavy settings with better  performance compared to recent baselines on tasks requiring long-context understanding.
>
> ---
> We hope that through our comments, we have been able to address the questions raised by the reviewer. In case the reviewer has any remaining, we are happy to address them.
>
> Best,

---

> > ### Comment · Reviewer_8B3g · 2025-08-03
> >
> > Thanks for your reply. I will maintain my score to accept this paper.

---

### Official Review · Reviewer_an8B · 2025-06-30

**Clarity:** 2
**Significance:** 2
**Originality:** 3
**Rating:** 5
**Confidence:** 3

**Summary:**

This work proposes Neural Block Linearization (NBL) as a method of replacing non-linear or complex layers / blocks in a deep learning architecture by the optimal linearization obtained via the SVD of the input-output data matrix over a calibration dataset. This method provides clearer error bounds compared to pruning or other methods for speeding up inference. In experiments on NBL replacing Self-Attention blocks in LLMs, the author can improve inference speeds at only slight performance degradation.

**Questions:**

- Why is the limitation that applying NBL to attention blocks removes the history not covered in your paper?

- Is there a possiblity to extend this to general random input vectors by just analyzing the QKV matrices and how they apply to random vectors? This might reduce the reliance on the calibration dataset and improve the bounds to general vectors beyond calibration "in-distribution".

- It would be interesting to analyze the vector norms of the layers in comparison to the residual vector norms. This could explain why later layers are more likely to be chosen.
Could you investigate those?

**Ethical Concerns:**

["NO or VERY MINOR ethics concerns only"]

**Final Justification:**

The experimental results show an improvement over competitive methods, and together with the included limitations section (that was missing before to fulfill all mandatory submission requirements) and potential future work, I want to recommend this work for acceptance.

**Limitations:**

The paper is missing a limitations section.

The paper does not cover the limitation of its method that for linearizing self-attention to a linear relationship between per-token input and output is reduced to a 1-convolution.

**Paper Formatting Concerns:**

The paper is missing a limitations section.

**Quality:**

3

**Strengths And Weaknesses:**

Strengths:
- Theoretical derivation, analysis and bounds
- No training needed
- Identification of target layers with minimal performance impact after linearization

Weaknesses:
- There is no history for linearized attention blocks, outputs depending on the past are approximated by a linear average over typical outputs for this input in the calibration dataset
- When put to extreme (i.e. replacing all self-attention layers) this method should evolve into a bi-gram-like model, that only takes into account the previous token
- Calibration dataset needs to cover the whole inference distribution - acts similar as a training dataset
- missing limitations section

---

> ### Author Rebuttal · Authors · 2025-07-30
>
> We thank the reviewer for the feedback. Below we address the points raised.
>
> ## Weaknesses:
>
> > W1. There is no history for linearized attention blocks, outputs depending on the past are approximated by a linear average over typical outputs for this input in the calibration dataset
>
> NBL does not replace all attention layers. It only replaces those that can be well-approximated by a learned linear mapping, while keeping the rest—especially the ones important for modeling complex or long-range dependencies—unchanged. This allows the model to retain much of its ability to handle context and history.
>
> The replaced layers use a token-wise linear transformation learned from real activation data. This transformation is applied independently to each token, so it doesn't average over history or remove contextual information. **Many original attention layers still remain and perform dynamic computations**.
>
> Our results (Tables 2–5) show that even after replacing multiple layers, the model still performs well on tasks that rely on history. This confirms that the remaining attention layers and the learned linear mappings together preserve important contextual reasoning.
>
> > W2. When put to extreme (i.e. replacing all self-attention layers) this method should evolve into a bi-gram-like model, that only takes into account the previous token
>
> While it is true that full replacement of all attention layers would limit the model’s expressiveness, this scenario is **not aligned with the goals or design of our methodology**. NBL is explicitly built around **selective replacement**, using canonical correlation scores to retain layers that are not linearly predictable and likely essential for modeling long-range or nonlinear dependencies.
>
> Moreover, even the substituted layers do not behave like static averaging as they apply token-wise learned linear transformations, fitted to real activation data. These retain per-token contextual information without collapsing the output into a local or averaged form.
>
> Therefore, *although full replacement is theoretically possible, it is neither practiced nor advocated in our method*. Our empirical results demonstrate that **selectively replacing even a few layers yields efficiency gains without collapsing into shallow behavior**, preserving reasoning ability across a wide range of tasks as evidenced by our strong empirical results, consistently outperforming recent pruning baselines in maintaining performance.
>
> > W3. Calibration dataset needs to cover the whole inference distribution - acts similar as a training dataset
>
> Thank you for the comment. We would like to clarify that the calibration dataset does not act like a training dataset. **Our method does not involve gradient computation, parameter updates, or task-specific supervision**. Instead, our method performs a lightweight, unsupervised forward pass over 256 samples to estimate CCA-based linear approximability. This process is scale-invariant, task-agnostic, and computationally efficient.
>
> As shown in our experiment results in Tables 2–5, we calibrate using a generic corpus (C4 dataset) yet evaluate on reasoning-heavy tasks like ARC and MMLU, which have distinct distributional properties. The strong performance in these settings indicates strong robustness to calibration–inference mismatch.
>
> To further analyze this, we conducted an explicit ablation (Appendix F.1) where we calibrate on either C4 or WikiText-2, then evaluate on both datasets. Below, we summarize the perplexity results from Tables 14 and 15:
>
> |Calibration Dataset|Method|C4↓(Llama-3.1-8B)|C4↓(Mistral-7B)|WikiText-2↓(Llama-3.1-8B)|WikiText-2↓(Mistral-7B)|
> |-|-|-|-|-|-|
> |WikiText-2|SliceGPT-25%|123.29|28.28|14.27|7.61|
> ||SLEB-8|20.53|15.07|16.70|10.23|
> ||Attn DROP-8|11.40|9.20|7.39|6.81|
> ||**Attn NBL-8**|**12.22**|**9.85**|**7.18**|**5.59**|
> |C4|SliceGPT-25%|25.41|13.07|74.33|12.11|
> ||SLEB-8|20.54|15.49|16.67|11.13|
> ||Attn DROP-8|11.35|9.20|7.48|6.81|
> ||**Attn NBL-8**|**11.37**|**9.08**|**9.57**|**7.33**|
>
> As shown above, Attn NBL exhibits minimal degradation across calibration sources (e.g., C4 eval: 12.22 → 11.37), in contrast to SliceGPT, which suffers from large mismatches (e.g., WikiText-2 eval: 14.27 → 74.33). This **confirms that NBL is robust to moderate calibration–inference distribution shifts**. In short, calibration is computationally efficient, task-agnostic, and significantly lighter than training, both in cost and in assumptions.
>
> > W4. missing limitations section.
>
> We would like to clarify that we **do include a limitations section in Appendix-I** of the paper. There, we explicitly discuss several key assumptions of our method. We will make this section more detailed and more visible in the revised version by moving it to the main text.
>
> ---
> ## Questions:
> > Q1. Why is the limitation that applying NBL to attention blocks removes the history not covered in your paper?
>
> We respectfully disagree that NBL removes access to history, as our method does not replace all attention layers. Instead, it applies selective substitution, where only the layers whose outputs are well-approximated linearly (as measured by CCA scores) are replaced. Layers that are critical for modeling long-range dependencies or complex reasoning are retained. This data-driven approach ensures that the model preserves its ability to attend over extended context when necessary.
>
> The strong performance of our method on tasks requiring multi-hop or long-context reasoning (e.g., ARC, MMLU) in Tables 2–5 supports this point. Nonetheless, we agree that if NBL were applied indiscriminately or too aggressively — especially replacing early attention layers — it could, in the limit, reduce the model's effective receptive field, resembling a bi-gram-like architecture that lacks full history access.
>
> While we already mention a related limitation in Appendix-I (*"One limitation is that certain nonlinear transformations in LLMs may not admit low-error linear approximations."*), we agree this point deserves more emphasis. We will update the limitations and also add a discussion in the conclusion of the main text to clarify that:
>
> - *"With NBL, we currently rely on cautious and selective layer replacement based on linear approximability. However, aggressive substitution across all attention layers may reduce the model’s ability to capture the history and long-range dependencies. Future work will explore ways to mitigate this, such as applying NBL at the head level or retaining critical heads to preserve contextual reasoning, thereby achieving a finer tradeoff between efficiency and capability."*
>
> > Q2. Is there a possiblity to extend this to general random input vectors by just analyzing the QKV matrices and how they apply to random vectors? This might reduce the reliance on the calibration dataset.
>
> This is an insightful direction. Analyzing the QKV structure directly, for example, through probing or synthetic inputs, could provide a layerwise estimate of sensitivity or redundancy. However, **such methods may miss important correlations or positional biases present in real data**. Our Canonical Correlation Anslysis (CCA)-based method captures data informed structure, which is critical to preserving fidelity. This situation is existent in may prior works, such as SLEB, SliceGPT and DROP which utilizes calibration dataset to accurately model the data distribution while compressing LLMs.
>
> That said, we agree that hybrid approaches combining theoretical analysis of QKV structure with data-driven calibration could further improve robustness and generalization beyond the calibration distribution. We consider this a promising area for future work. We will add this point to the conclusion section as a potential direction for future work.
>
> - *"An interesting direction for future work is to combine theoretical analysis of QKV structure with data-driven calibration, enabling layer substitution decisions that generalize beyond the calibration distribution and reduce reliance on task-specific data."*
>
> > Q3. It would be interesting to analyze the vector norms of the layers in comparison to the residual vector norms. This could explain why later layers are more likely to be chosen.
>
> Thank you for the suggestion. While norm-based diagnostics may offer interpretability, our method is not driven by absolute magnitudes. Instead, we use **Normalized Mean Squared Error (NMSE)** to evaluate how well the output of each attention layer can be linearly approximated **- independent of scale**. Therefore, raw output or residual norms alone do not determine which layers are substituted. Importantly, **our focus is on preserving functional computation paths, not on the magnitude of activations or weights.**
>
> But, we investigated the ratio $\|Y\| / \|X + Y\|$ of attention layer output's ($Y$) norm to the residual's ($X + Y$) norm, in DeepSeek-8B. However, this analysis did not reveal any meaningful correlation with the layers chosen by our method. The top-4 layers ranked by $\|Y\| / \|X + Y\|$ are:
>
> - Layer indices with highest $\|Y\| / \|X + Y\|$: `[0, 1, 31, 30]`
>
> This further supports our choice to use **NMSE**, which is scale-invariant and more directly reflects functional compressibility. Parameter norms can also be analyzed, but they are entangled with output norms, being hard to interpret.  That said, **if the reviewer has further details for a more specific diagnostic or hypothesis in mind regarding the role of vector norms, we would be happy to explore it during the discussion period.**
>
> ---
> ## Limitations
> > L1. The paper is missing a limitations section ... the paper does not cover the limitation of its method ...
>
> We already included a limiations section in Appendix-I. However, we will extend this and move it to the main text as pointed by the reviewer. Please see Weakness-4 and Question-1 for further details.
>
> ---
> We hope our responses address the points raised by the reviewer and are happy to clarify any remaining.
>
> Best,

---

> > ### Comment · Reviewer_an8B · 2025-08-02
> >
> > Thank you for all your clarifications.
> >
> > Regarding [W3], the mentioned phrase in the appendix strongly falls short of a limitations discussion. These should include (in more detail as I discussed here) the problem of "averaging the history" on the calibration dataset for the attention layers, potentially leading to hallucination-like effects (predicting something that is commonly present in the attention context, but which isn't actually there in inference). The mentioned limitation of losing all history, when replacing all attention layers. The now verified impact of the calibration dataset (which is admittedly small but present). All of this should be presented in a dedicated section in the main paper.
> >
> > Regarding Q2, cool that you find this direction interesting as well.
> >
> > Regarding Q3, using the NMSE is probably a good choice for evaluating the impact of linearization of a single layer, but maybe a re-weighted MSE (by the $|Y| / |X + Y|$ ratio) might even give a better choice for the overall impact in the whole network. Maybe this could be mentioned for potential future work as well.

---

> ### Author Response · Authors · 2025-08-03
> **Response to Reviewer an8B's Comments**
>
> We thank the reviewer for the constructive feedback. In particular, we have created a new section titled *“Further Discussions, Possible Limitations, and Future Work”*, which consolidates and expands on the points you raised.
>
> A key contribution in our work is to identify blocks that can be simplified with their optimal LMMSE approximation mantaining accuracy while reducing computational costs. We replace attention layers *selectively* based on our canonical correlation bound. We do *not* aim to replace all layers; rather, we aim to identify a *subset* of layers that are most suitable to be accelerated.
>
> **Moreover, you mentioned “the now verified impact of the calibration dataset (which is admittedly small but present)”. We would like to clarify that this impact was already analyzed in our original submission through the ablation study in *Appendix F.1*, where we evaluated calibration on both C4 and WikiText-2 and tested generalization across datasets.** As shown in Tables 14 and 15, **NBL shows stable performance across across calibration sources**, especially compared to previous methods like SliceGPT. We agree this discussion should be brought more prominently, which we will add in main text with additional page, also discussing room for further improvement.
>
> Additionally, we will explicitly address potential limitations, including risks of reduced history modeling if attention layers are substituted indiscriminately and hallucination-like effects due to, e.g., insufficient calibration coverage. While these effects have not impacted reasoning performance in our experiments, we now state them clearly as possible limit failure modes and we discuss possible mitigations. We have also expanded the future work component, including QKV structure analysis, operator norms, and hybrid approaches such as head-level substitution.
>
> In summary, the main focus of this work is to use canonical correlation analysis as selection criterion for LMMSE estimators minimizing approximation error. Extensive experiments show minimal accuracy drop not only in reasoning tasks but also in attention-heavy tasks such as long-context tasks and RULER (Q2 of Rev. 8B3g). Overall, NBL shows a better accuracy-speed trade-off than baselines and opens promising directions for future work in LLM inference optimization.
>
> We hope the **following new section below** (that will be incorporated to the main text), fully addresses your comments and thanks again for helping improve the clarity of the paper.
>
> ---
> ## New Section: *Further Discussions, Possible Limitations, and Future Work*.
> *Empirical results show the effectiveness of our approach across several challenging reasoning benchmarks (e.g., MMLU, HellaSwag, ARC). It is further useful to note some possible limitations. In general, certain nonlinear transformations in large language models may not admit accurate linear approximations. This is an inherent property of deep architectures. NBL addresses this by ranking layers based on linear predictability via canonical correlation analysis and substituting only those with low approximation error. Our method is explicitly designed around selective replacement; an aggressive substitution of all attention layers, which would reduce the model to shallow, history-agnostic behavior, is not suggested by our approach.*
>
> *The choice of calibration dataset can affect performance on new inputs, where a form of calibration history averaging could lead to hallucination-like effects. These effects could be possible especially in domain shift or insufficient calibration coverage cases. Our ablations (Appendix F.1) show that NBL demonstrates superior robustness than SliceGPT and that NBL's performance is remarkably stable even in the case of two very different calibration distributions (C4 and WikiText-2). Additionally, although calibrated on C4, we tested on reasoning tasks with possibly very disparate data distributions, yielding successful results (Tables 2-5). Future directions for possible mitigations to calibration data dependency are given below.*
>
> *We believe NBL offers significant promise for future research on optimizing LLM inference. Although NBL shows only mild sensitivity to calibration data, its limited dependence on inputs can be further reduced using calibration data mixture techniques or adaptive strategies. A promising direction is to combine theoretical analysis of QKV structure with data-driven calibration to improve generalization and reduce reliance on task-specific data. More principled selection of layers could also be achieved by analysing operator norms of attention layers, or examining the internal behavior of attention matrices. Investigating the relative norms of attention outputs and residual connections may further complement the CCA based criterion. Finally, hybrid designs that retain critical attention heads within a layer present a promising path for balancing inference efficiency with contextual reasoning capacity.*
>
> ---
> Best,

---

> > ### Comment · Reviewer_an8B · 2025-08-03
> >
> > Thank you for presenting the detailed limitations and future work section you are planning to integrate. This alleviates my main structural point of concern.
> > Beyond this I believe the paper contains a good idea with solid experiments, so I would recommend it for acceptance now.

---

### Official Review · Reviewer_mjoZ · 2025-07-03

**Clarity:** 3
**Significance:** 3
**Originality:** 3
**Rating:** 4
**Confidence:** 4

**Summary:**

The authors introduce a fine-tuning-free method that uses Canonical Correlation Analysis (CCA) to identify and replace attention blocks in pretrained LLMs with their best linear approximations. This approach significantly reduces computational costs while largely preserving the model's original performance. The key innovation is identifying which attention blocks behave most like linear layers and can therefore be simplified without extensive retraining.

**Questions:**

Already listed in the Weaknesses

**Ethical Concerns:**

["NO or VERY MINOR ethics concerns only"]

**Final Justification:**

The CCA based linearization method itself is of interest and the authors provide rather comprehensive experimental studies.

**Quality:**

3

**Strengths And Weaknesses:**

Strengths:

1. The paper is generally well-written and the motivations and methods are clearly stated. Necessary theoretical analysis is provided.
2. The authors conduct rather complete evaluation with ablation study, demonstrates the strong performance of the proposed method.

Weaknesses:

1. I'm wondering what is the purpose of using the upper bound of Theorem 3.2 in the algorithm rather than directly evaluating the NMSE by definition. To obtain the sum of $\rho_i$, you need to compute $C_W$ already (or the trace of $C_W$). By a simple expansion of the MSE, it is not hard to see that
$$\mathrm{MSE} = \mathrm{Tr}[C_{YY} - C_{YX} C_{XX}^{-1} C_{YX}^\top].$$
It seems calculating both objectives have roughly the same cost. I don't see the necessity of using the approximation. I'm happy for further discussion on this point.
2. Is it a typo? In the algorithm output line 11: "Compressed LLM with linearized attention layers." -> linear layer actually.

---

> ### Author Rebuttal · Authors · 2025-07-30
>
> We thank the reviewer for the feedback. Below we address the points raised.
>
> ---
> ## Strengths:
> > The paper is generally well-written and the motivations and methods are clearly stated. Necessary theoretical analysis is provided.
>
> > The authors conduct rather complete evaluation with ablation study, demonstrates the strong performance of the proposed method.
>
> We appreciate the positive comments by the reviewer.
>
> ---
> ## Weaknesses:
>
> >  I'm wondering what is the purpose of using the upper bound of Theorem 3.2 in the algorithm rather than directly evaluating the NMSE by definition. To obtain the sum of, you need to compute C_{W} already (or the trace of C_{W}). By a simple expansion of the MSE, it is not hard to see that $\mathrm{MSE} = \mathrm{Tr}[C_{YY} - C_{YX} C_{XX}^{-1} C_{YX}^\top]$. It seems calculating both objectives have roughly the same cost. I don't see the necessity of using the approximation. I'm happy for further discussion on this point.
>
> We appreciate the reviewer’s thoughtful question and are happy to clarify our rationale. While both the NMSE and the upper bound in Theorem 3.2 rely on second-order statistics (i.e., covariance matrices), we intentionally use the CCA-based bound instead of the direct NMSE expression for both theoretical and practical reasons.
>
> The scalar NMSE, defined as:
> - $\mathrm{NMSE} = \frac{\mathrm{Tr}[C_{YY} - C_{YX} C_{XX}^{-1} C_{YX}^\top]}{\mathrm{Tr}[C_{YY}]}$,
>
> which aggregates the approximation quality into a single number, obscures the contribution of individual directions in the output space. In contrast, the bound in Theorem 3.2 decomposes the error along canonical directions via
> - $\mathrm{NMSE} \leq (h_{\text{out}} - r) + \sum_{i=1}^r (1 - \rho_i^2)$,
>
> where $\rho_i$ are the canonical correlations.
>
> This decomposition allows us to have a **whitened** version of the NMSE enabling the understanding of *how many* directions are well-approximated and *how strongly* each output mode aligns with the input, which NMSE does not offer. Therefore, although MSE and NMSE is a commonly used criterion in the estimation theory literature [1,2] (*and we aimed to preserve this connection through our bound*), directly using the NMSE can be biased by high-variance directions, masking poor approximations.
>
> Because the NMSE normalizes by $\mathrm{Tr}[C_{YY}]$, it can be dominated by a small number of high-variance output directions. Even if the approximation fails in lower-variance (but semantically important) dimensions, NMSE can still appear deceptively low. In contrast, the CCA-based bound evaluates approximation error uniformly across canonical modes, without being skewed by marginal variances.
>
> Therefore, our method is not just trying to minimize average error, but to **identify layers whose outputs are largely determined by their inputs in a linear fashion**. Canonical correlations directly quantify this, where they are the optimal linear correlations between input and output projections. This makes them an ideal indicator for substitutability.
>
> To assess our claim, we performed an ablation where we calculated the layer rankings using both the direct NMSE and the CCA based criterion on the Mistral-7B model as the following:
>
> - NMSE values sorted from low to high: `[2, 3, 4, 5, 6, 7, 8, 9, 10, 11, 12, 25, 22, 26, 24, 13, 27, 14, 23, 28, 21, 16, 20, 15, 30, 31, 17, 19, 18, 29, 1, 0]`
>
> - CCA criterion values sorted from low to high: `[25, 26, 27, 24, 22, 28, 23, 30, 31, 21, 29, 20, 19, 18, 17, 16, 6, 8, 9, 12, 14, 13, 11, 15, 4, 10, 7, 5, 3, 0, 2, 1]`
>
> Then, replacing 12 attention layers with the CCA criterion in the Mistral-7B model drops the average reasoning accuracy of the model **70.2% $\rightarrow$ 68.3%** as also indicated in Table-2 of our paper, whereas **the replacement of 12 layers with the direct NMSE causes a much larger performance drop of 70.2% $\rightarrow$ 39.1%**, suggesting the better performance of the CCA based criterion.
>
> Furthermore, we also would like to note that while both approaches (direct NMSE vs CCA based bound) involve similar matrix operations (inversions and matrix multiplications vs. whitening and SVD), we found the CCA route more stable in practice, especially in low-data or near-singular regimes. The whitening step in CCA ensures that numerical instabilities in covariance estimation do not overly bias the error estimate.
>
> To sum up, our framework is built around Canonical Correlation Analysis as a network analysis tool [3,4] to understand and exploit representational redundancy.
>
> References:
> - [1]. Steven M Kay. Fundamentals of statistical signal processing: Estimation theory, 1993.
> - [2]. Thomas Kailath, Ali H Sayed, and Babak Hassibi. Linear estimation. Prentice Hall, 2000.
> - [3]. Maithra Raghu et al. "Svcca: Singular vector canonical correlation analysis for deep learning dynamics and interpretability", Neurips, 2017.
> - [4]. Ari Morcos et al. "Insights on representational similarity in neural networks with canonical correlation", Neurips, 2018.
>
> ---
>
> > Is it a typo? In the algorithm output line 11: "Compressed LLM with linearized attention layers." -> linear layer actually.
>
> Thank you for pointing this out. We agree that the phrasing could be improved. Our intention was to convey that the selected attention layers are replaced with their corresponding optimal linear approximations, each implemented as a single linear layer. We will revise the wording in Algorithm 1, line 11 to more clearly state: *"Compressed LLM with selected attention layers replaced by linear layers."*
>
> ---
> We hope that through our comments, we have been able to address the questions raised by the reviewer. In case the reviewer has any remaining questions, we are happy to address them.
>
> Best,

---

> > ### Comment · Reviewer_mjoZ · 2025-08-03
> >
> > Thanks to the author for the clarification on the design choice between CCA and NMSE, please ensure this is highlighted in the revison. My evaluation of the work remains unchanged.

---

### Note · Authors · 2025-08-11

Dear Program Chairs, Area Chairs, and Reviewers,

We thank all reviewers for their constructive and engaged feedback.

---
**Reviewer recognition:** All reviewers agree on the paper’s strong theoretical foundation, clarity, and comprehensive experiments.

> - **mjoZ**: *“The authors conduct rather complete evaluation with ablation study, demonstrates the strong performance of the proposed method.”*, after clarifications on CCA and NMSE, confirmed positive evaluation.
> - **an8B**: *“Good idea with solid experiments… would recommend it for acceptance now.”*, after we integrated an expanded limitations/future work section in the main text.
> - **8B3g**: *“Clear theoretical foundation; extensive experiments show the effectiveness of NBL on reasoning tasks... I will maintain my score to accept this paper. ”*,  after clarifications on context length and long-range dependencies.
> - **ZrVu**: *“Thanks for the additional baselines… better position the work.”*, increased score after we added further empirical evidence.

---
**Main points of rebuttal**: Below we summarize the key points addressed during the discussions (full details in individual responses):

- **Expanded baselines & metrics:** Added PruneNET, SliceGPT, and SLEB results for all models; reported explicit sparsity levels; extended tables for DeepSeek-8B and Llama-3.1-70B. Further demonstrated benefits of NBL on attention heavy tasks.
- **Efficiency vs pruning baselines:** Demonstrated that NBL offers superior accuracy–speed and accuracy–KV-cache trade-offs compared to recent pruning baselines, especially in mid/high compression regimes, with small calibration cost.
- **CCA vs. NMSE/cosine dist.**: Added discussions and empirical studies motivating using CCA as a principled criterion to select layers most suitable to be accelerated, capturing input-ouput structural alignments that other metrics may overlook.
- **Scope, limitations & future work**: Added a new limitations/discussion section to the main text. Clarified that NBL performs selective replacement of layers, focusing on attention layers to preserve long-range dependencies. Discussed possible risks and outlined future works.

---
We will revise the paper to incorporate all reviewers' suggestions, expanded tables, and the improved limitations/future work discussion, ensuring the final version reflects the clarifications made during the review process.

Best regards,

Authors

---

### Decision · Program_Chairs · 2025-09-17

**Decision:**

Accept (poster)

**Comment:**

This paper proposes a technique for linearizing self-attention layers in Transformers to achieve faster inference, using a linear minimum mean squared error estimator. The reviewers felt that the approach was novel, with strong theoretical backing and demonstrated wall-clock speedup, and solid experiments. During the rebuttal period, the authors responded well to reviewer questions and reported additional experiments that made a stronger case for the method.

I recommend accept as a poster.